# NEURAL SYSTEMATIC BINDER

**Gautam Singh**[1]*, **Yeongbin Kim**[2] **& Sungjin Ahn**[2]
[1]Rutgers University
[2]KAIST

## ABSTRACT

The key to high-level cognition is believed to be the ability to systematically manipulate and compose knowledge pieces. While token-like structured knowledge representations are naturally provided in text, it is elusive how to obtain them for unstructured modalities such as scene images. In this paper, we propose a neural mechanism called *Neural Systematic Binder* or *SysBinder* for constructing a novel structured representation called *Block-Slot Representation*. In Block-Slot Representation, object-centric representations known as slots are constructed by composing a set of independent factor representations called *blocks*, to facilitate systematic generalization. SysBinder obtains this structure in an unsupervised way by alternatingly applying two different binding principles: spatial binding for spatial modularity across the full scene and factor binding for factor modularity within an object. SysBinder is a simple, deterministic, and general-purpose layer that can be applied as a drop-in module in any arbitrary neural network and on any modality. In experiments, we find that SysBinder provides significantly better factor disentanglement within the slots than the conventional object-centric methods, including, for the first time, in visually complex scene images such as CLEVR-Tex. Furthermore, we demonstrate factor-level systematicity in controlled scene generation by decoding unseen factor combinations. `https://sites.google.com/view/neural-systematic-binder`

## 1 INTRODUCTION

One of the most remarkable traits of human intelligence is the ability to generalize systematically. Humans are good at dealing with out-of-distribution situations because of the ability to understand them as a composition of pre-acquired knowledge pieces and knowing how to manipulate these pieces (Lake et al., 2017; Fodor & Pylyshyn, 1988). This also seems to be foundational for a broad range of higher-level cognitive functions, such as planning, reasoning, analogy-making, and causal inference. However, realizing this ability in machines still remains a major challenge in modern machine learning (McVee et al., 2005; Bottou, 2014; Schölkopf et al., 2021).

This problem is considerably more difficult for unstructured modalities such as visual scenes or speech signals compared to modalities such as language. In language, we can consider embeddings of words or other forms of tokens as modular knowledge pieces. However, for unstructured modalities, we would first need to obtain such tokens, e.g., by grouping relevant low-level features, through a process called *binding* (Greff et al., 2020a). Yet it is quite elusive what should be the appropriate structure and granularity of these tokens to support systematic generalization and how to obtain them, particularly in the unsupervised setting where the model should learn this ability only by observing.

In visual scenes, binding has recently been pursued by object-centric learning methods through the *spatial binding* approach (Locatello et al., 2020; Singh et al., 2022a). Spatial binding aims to divide a scene spatially into smaller areas so that each area contains a meaningful entity like an object. The information in each area is then grouped and aggregated to produce a representation of an object i.e., a slot, resulting in a set of slots per scene. These slots can be seen as independent modular knowledge pieces describing the full scene. However, the main limitation of the current methods based on spatial binding is that a slot is an entangled vector and not a composition of independent

---

*Correspondence to `singh.gautam@rutgers.edu` and `sjn.ahn@gmail.com`.

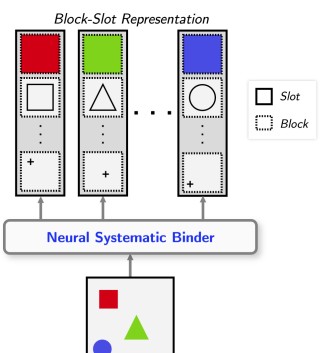 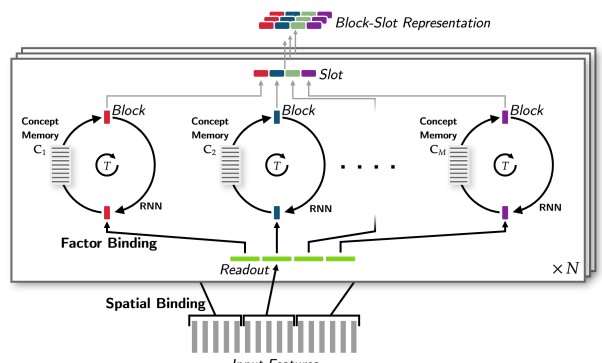

Figure 1: **Overview. Left:** We propose a novel binding mechanism, *Neural Systematic Binder*, that represents an object as a *slot* constructed by concatenating multi-dimensional factor representations called *blocks*. Without any supervision, each block learns to represent a specific factor of the object such as color, shape, or position. **Right:** Neural Systematic Binder works by combining two binding principles: spatial binding and factor binding. In *spatial binding*, the slots undergo a competition for each input feature followed by iterative refinement, similar to Slot Attention. In *factor binding*, unlike Slot Attention, for each slot, the bottom-up information from the attended input features is split and routed to $M$ independent block refinement pathways. Importantly, each pathway provides a representation bottleneck by performing dot-product attention on a memory of learned prototypes.

modular representations. As such, a systematically novel object would map to an unfamiliar slot vector rather than a modular combination of familiar factor tokens, such as color, shape, and position.

In this paper, we propose the *Neural Systematic Binder*, or *SysBinder* for short. SysBinder generalizes the conventional binding process by combining *spatial binding* and *factor binding*. While spatial binding provides spatial modularity across the full scene, factor binding provides factor modularity within an object. More specifically, given an input, SysBinder produces a set of vector representations, called slots, via spatial binding. However, unlike typical object-centric learning, each of these slots is constructed by concatenating a set of independent factor representations called blocks obtained via factor binding. SysBinder uses iterative refinement to learn the slots by applying spatial binding step and factor-binding steps alternatingly during the refinement process. In the spatial binding step, slots compete with each other to find an input attention area per slot. The information in each area is then grouped and used to refine the blocks. Crucially, to achieve factor binding, each block is refined in a modular fashion by applying an independent RNN and a soft information bottleneck on each block. We train SysBinder in a fully unsupervised manner by reconstructing the input from the slots using the decoding framework of SLATE (Singh et al., 2022a;b).

The contributions of this paper are as follows. First, SysBinder is the first deterministic binding mechanism to produce disentangled factors within a slot. This deterministic nature of SysBinder is remarkable since, conventionally, probabilistic modeling has been considered crucial for the emergence of factors within slots (Higgins et al., 2017; Greff et al., 2019). Second, in probabilistic frameworks, the representation of a factor is a single dimension of the slot, while in SysBinder, the representation of a factor is a multi-dimensional block, providing a more flexible and richer way to represent a factor. Third, similar to Slot Attention, SysBinder is a simple, deterministic, and general-purpose layer that can be applied as a drop-in module in any arbitrary neural network and on any modality. In experiments, we show that *i)* SysBinder achieves significantly better factor disentanglement within the slots than conventional object-centric methods, including those based on probabilistic frameworks. *ii)* Notably, for the first time in this line, we show factor emergence in visually complex scene images such as CLEVR-Tex (Karazija et al., 2021b). *iii)* Using the emergent factor blocks obtained from SysBinder, we demonstrate factor-level systematicity in controlled scene generation by decoding unseen block combinations. *iv)* Lastly, we provide an extensive analysis by evaluating the performance of several key model variants.

## 2 METHOD

### 2.1 BLOCK-SLOT REPRESENTATION

We propose and define *Block-Slot Representation* as a set of slots where each slot is constructed by concatenating a set of factor representations called *blocks*. For images, a slot would represent one object in the scene, similar to the conventional slots. However, unlike the conventional slots which either *i)* do not seek intra-slot factors at all (Locatello et al., 2020) or *ii)* seek to make each individual dimension a factor (Greff et al., 2019), we seek to make each multi-dimensional block a factor. Each block should represent at most one factor of the object, and, conversely, each factor should be fully represented by a single block. Formally, by denoting the number of slots by $N$, the number of blocks per slot by $M$, and the dimension of a block by $d$, we denote the collection of $N$ slots by a matrix $\mathbf{S} \in \mathbb{R}^{N \times Md}$. Each slot $\mathbf{s}_n \in \mathbb{R}^{Md}$ is a concatenation of $M$ blocks. We denote the $m$-th block of slot $n$ as $\mathbf{s}_{n,m} \in \mathbb{R}^d$.

### 2.2 SYSBINDER: NEURAL SYSTEMATIC BINDER

We propose *Neural Systematic Binder* or simply *SysBinder* as a novel binding mechanism to encode a set of $L$ input vectors $\mathbf{E}$ into Block-Slot Representation $\mathbf{S}$. SysBinder is a general-purpose mechanism that, in principle, can be used for any input modality (e.g., speech signals) as long as it is provided as a set of vectors. It can therefore be seen as a drop-in module that can be plugged into any arbitrary neural architecture. SysBinder is based on two binding principles: *spatial binding* and *factor binding*. In spatial binding, each slot $\mathbf{s}_n$ performs competitive attention on the input features $\mathbf{E}$ to find a spatial attention area. In factor binding, each of the $M$ blocks is updated modularly using the information from the attended spatial area by applying an independent RNN and a soft information bottleneck on each block. We first initialize the slots by randomly sampling them from a learned Gaussian distribution. We then refine the slots by alternatingly applying the spatial binding and factor binding steps multiple times. We now describe a single refinement iteration in detail.

#### 2.2.1 SPATIAL BINDING

In the spatial binding, the slots $\mathbf{S} \in \mathbb{R}^{N \times Md}$ first perform competitive attention (Locatello et al., 2020) on the input features $\mathbf{E} \in \mathbb{R}^{L \times D}$ to attend to a local area. For this, we apply linear projection $q$ on the slots to obtain the queries, and projections $k$ and $v$ on the inputs to obtain the keys and the values, all having the same size $Md$. We then perform dot-product between the queries and the keys to get the attention matrix $\mathbf{A} \in \mathbb{R}^{N \times L}$. In $\mathbf{A}$, each entry $\mathbf{A}_{n,l}$ is the attention weight of slot $n$ for attending over the input vector $l$. We normalize $\mathbf{A}$ by applying softmax across slots i.e., along the axis $N$. This implements a form of competition among slots for attending to each input $l$.

We now seek to group and aggregate the attended inputs and obtain the attention readout for each slot. For this, we normalize the attention matrix $\mathbf{A} \in \mathbb{R}^{N \times L}$ along the axis $L$, and multiply it with the input values $v(\mathbf{E}) \in \mathbb{R}^{L \times Md}$. This produces the attention readout in the form of a matrix $\mathbf{U} \in \mathbb{R}^{N \times Md}$ where each row $\mathbf{u}_n \in \mathbb{R}^{Md}$ is the readout corresponding to slot $n$.

$$\mathbf{A} = \operatorname*{softmax}_{N}\left(\frac{q(\mathbf{S}) \cdot k(\mathbf{E})^T}{\sqrt{Md}}\right) \quad \Longrightarrow \quad \mathbf{A}_{n,l} = \frac{\mathbf{A}_{n,l}}{\sum_{l=1}^{L} \mathbf{A}_{n,l}} \quad \Longrightarrow \quad \mathbf{U} = \mathbf{A} \cdot v(\mathbf{E}),$$

#### 2.2.2 FACTOR BINDING

In factor binding, we seek to use the readout information obtained from spatial binding and update each block in a modular manner. This is achieved via the following two steps:

**Modular Block Refinement.** For this, we split each slot's readout vector $\mathbf{u}_n$ into $M$ vector segments of equal sizes (denoting the $m$-th segment as $\mathbf{u}_{n,m} \in \mathbb{R}^d$). We then apply a refinement RNN independently on each block $\mathbf{s}_{n,m}$ in parallel by providing its corresponding readout segment $\mathbf{u}_{n,m}$ as the input. The RNN is implemented using a $\mathtt{GRU}_{\phi_m}$ and a residual $\mathtt{MLP}_{\phi_m}$. In our design, we choose to maintain separate parameters $\phi_m$ for each $m$ as this design can potentially scale better for complex large-scale problems where the number of blocks can be large. However, depending on the context, it is also a sensible design to share the parameters $\phi_m = \phi$ for all $m$.

$$\mathbf{s}_{n,m} = \mathtt{GRU}_{\phi_m}(\mathbf{s}_{n,m}, \mathbf{u}_{n,m}) \quad \Longrightarrow \quad \mathbf{s}_{n,m} \mathrel{+}= \mathtt{MLP}_{\phi_m}(\mathtt{LN}(\mathbf{s}_{n,m})).$$

**Block Bottleneck.** Next, we apply a soft information bottleneck on each block by making each block retrieve a representation from a concept memory. A concept memory $\mathbf{C}_m \in \mathbb{R}^{K \times d}$ is a set of $K$ learnable vectors or $K$ latent *prototype* vectors associated with the particular factor $m$. Each block $\mathbf{s}_{n,m}$ performs dot-product attention on its corresponding concept memory $\mathbf{C}_m$, and the retrieved vector is used as the block representation:

$$\mathbf{s}_{n,m} = \left[ \operatorname*{softmax}_K \left( \frac{\mathbf{s}_{n,m} \cdot \mathbf{C}_m^T}{\sqrt{d}} \right) \right] \cdot \mathbf{C}_m.$$

The spatial binding and the factor binding together form one refinement iteration. The refinement iterations are repeated several times and the slots obtained from the last iteration are taken to be the final Block-Slot Representation for providing downstream. We provide a pseudo-code of our method in Algorithm 1.

## 2.3 UNSUPERVISED OBJECT-CENTRIC LEARNING WITH SYSBINDER

In this section, we propose an unsupervised object-centric learning method for images using the proposed binding mechanism, SysBinder. For this, we adopt an auto-encoding approach in which we apply SysBinder to encode the image and then apply an autoregressive transformer decoder to reconstruct the image.

**Encoding.** Given an image $\mathbf{x} \in \mathbb{R}^{H \times W \times C}$, we first obtain a set of image features using a backbone encoder network, e.g., a CNN in our experiments. In this CNN, positional encodings are added in the second-last layer and the final output feature map is flattened to form a set of input vectors represented in a matrix form $\mathbf{E} \in \mathbb{R}^{L \times D}$. We then apply SysBinder on $\mathbf{E}$ to obtain slots $\mathbf{S}$.

**Block Coupling.** One crucial benefit realized by Block-Slot Representation is that the relationships and interactions among the representations can be modeled explicitly at the level of blocks rather than the level of slots. To obtain this benefit, we would like to provide all block vectors together as a set of tokens $\{\mathbf{s}_{n,m}\}$ to the downstream transformer decoder. However, because a set is order-less, it is difficult for the transformer *(i)* to know which concept each block is representing and *(ii)* to identify to which slot each block is belonging. Following the traditional approach for dealing with the order issue in Transformer, one possible approach to this problem is to provide two positional embeddings to each block indexed by $(n, m)$: one for slot membership $n$ and the other for the block index $m$. However, this would mean that the decoder should take the burden of learning invariance to the slot order which is not built-in into the design. Furthermore, it could be difficult for the decoder to generalize well when many more slots than the number seen during training are given at test time.

To avoid these challenges, we propose the following method. To each block, we add a positional encoding $\mathbf{p}_m^{\text{block}}$ that depends only on $m$ but not on $n$. We then let the blocks belonging to the same slot interact via a 1-layer transformer, termed *block coupler*, as follows:

$$\bar{\mathbf{s}}_{n,m} = \mathbf{s}_{n,m} + \mathbf{p}_m^{\text{block}} \qquad \Longrightarrow \qquad \tilde{\mathbf{s}}_{n,1}, \ldots, \tilde{\mathbf{s}}_{n,M} = \text{BlockCoupler}(\bar{\mathbf{s}}_{n,1}, \ldots, \bar{\mathbf{s}}_{n,M})$$

After this, we provide all the returned vectors to the transformer decoder as a single set $\mathcal{S} = \{\tilde{\mathbf{s}}_{n,m}\}$. With this approach, the decoder is informed about the role of each block via the positional encoding $\mathbf{p}_m^{\text{block}}$. Because the blocks in the same slot have interacted with each other, the information about which blocks belong to the same slot is also implicitly available to the decoder. Furthermore, the decoder treats the slots as order-invariant and can also deal with more slots than shown in training.

**Autoregressive Decoding.** Conditioned on $\mathcal{S}$ obtained via block coupling, a transformer decoder learns to reconstruct the image. Similarly to the decoder of SLATE (Singh et al., 2022a) and STEVE (Singh et al., 2022b), the transformer does not directly reconstruct the pixels of the image. Instead, we reconstruct a discrete code representation $\mathbf{z}_1, \ldots, \mathbf{z}_{L'}$ of the given image $\mathbf{x}$ obtained from a discrete VAE (dVAE). In autoregressive decoding, the transformer predicts each code $\mathbf{z}_l$ at position $l$ by looking at all the previous codes $\mathbf{z}_1, \ldots, \mathbf{z}_{l-1}$ and the input blocks $\mathcal{S}$. For this, we first retrieve a learned embedding for each discrete code $\mathbf{z}_1, \ldots, \mathbf{z}_{l-1}$. To these, we add positional encodings and provide the resulting embeddings $\mathbf{e}_1, \ldots, \mathbf{e}_{l-1}$ to the transformer to predict the next token:

$$\mathbf{e}_l = \text{Dictionary}_\theta(\mathbf{z}_l) + \mathbf{p}_l^{\text{token}} \qquad \Longrightarrow \qquad \mathbf{o}_l = \text{TransformerDecoder}_\theta(\mathbf{e}_1, \ldots, \mathbf{e}_{l-1}; \mathcal{S}),$$

where $\mathbf{o}_l$ are the predicted logits for the token at position $l$. The training is done by learning to predict all tokens in parallel using causal masking via the learning objective $\mathcal{L} = \sum_{l=1}^{L'} \text{CrossEntropy}(\mathbf{z}_l, \mathbf{o}_l)$. The dVAE may be pre-trained or trained jointly with the model. For more details, see Appendix D.4.

## 3 EVALUATING BLOCK-SLOT REPRESENTATION

In this section, we introduce the method to quantitatively evaluate Block-Slot Representations based on the DCI framework (Eastwood & Williams, 2018). The DCI framework is multi-faceted and evaluates multiple aspects of the quality of a representation vector in terms of Disentanglement (D), Completeness (C), and Informativeness (I). Although the original DCI framework is designed for a single vector representation, Nanbo et al. (2020) proposed a method to make it applicable to a set of slots by first matching the slots with the true objects using IoU before applying the DCI procedure.

Specifically, let $\mathcal{S} = (\mathbf{s}_1, \ldots, \mathbf{s}_O)$ be a set of slots, with a slot $\mathbf{s}_o \in \mathbb{R}^d$. Also, for each object, let there be $K$ ground-truth factors of object variation such as color, shape, and position. We train $K$ probe functions $g_1^{\text{probe}}, \ldots, g_K^{\text{probe}}$, one for each factor, to predict the factor label $y_o^k$ from the slot vector $\mathbf{s}_o$. The probes are classifiers implemented using gradient-boosted trees (Locatello et al., 2019). Using the trained probes, we can obtain a feature importance matrix $\mathbf{R} = (R_{k,j}) \in \mathbb{R}^{K \times d}$. Each entry $R_{k,j}$ denotes the importance of feature dimension $j$ in predicting the factor $k$ and is obtained from the corresponding probe $g_k^{\text{probe}}$. Using the importance matrix, the disentanglement-score $\mathtt{D}_j$ for a specific feature dimension $j$ and the completeness-score $\mathtt{C}_k$ for a specific factor $k$ are computed as follows:

$$\mathtt{D}_j = 1 - H_K(R_{:,j}), \qquad\qquad \mathtt{C}_k = 1 - H_d(R_{k,:}),$$

where $H_K(\cdot)$ denotes entropy with log-base $K$ and $H_d(\cdot)$ denotes entropy with log-base $d$. From these, the final disentanglement-score $\mathtt{D}$ and completeness-score $\mathtt{C}$ are computed as the weighted average of $\mathtt{D}_j$ over all the dimensions $j = 1, \ldots, d$ and similarly for $\mathtt{C}_k$ by averaging over $k = 1, \ldots, K$. The informativeness-score is the average of the prediction accuracies of all the probes.

For evaluating the Block-Slot Representation, we first apply the same probing approach as described above to obtain the importance matrix $\mathbf{R} = (R_{k,j}) \in \mathbb{R}^{K \times Md}$, where $M$ is the number of blocks, $d$ is the block size, and $Md$ is the size of the slot. To evaluate the representation in the setting where each block is an independent factor, we obtain an importance score per block by summing the importance scores across the dimensions belonging to the same block, giving us block importance matrix $\mathbf{R}^{\text{block}} = (R_{k,m}^{\text{block}}) \in \mathbb{R}^{K \times M}$. Using these block importance values, we compute the disentanglement-scores and the completeness-scores in the same way as described above.

## 4 RELATED WORK

Our work belongs in the line of unsupervised object-centric learning methods (Greff et al., 2016; Burgess et al., 2019; Greff et al., 2019; Locatello et al., 2020; Greff et al., 2017; Engelcke et al., 2019; Engelcke et al., 2021; Anciukevicius et al., 2020; von Kügelgen et al., 2020; Greff et al., 2020b; Singh et al., 2022a; Chang et al., 2022). In this line, intra-slot disentanglement has been explored inside the VAE framework (Greff et al., 2019; Burgess et al., 2019; Nanbo et al., 2020; Zoran et al., 2021). However, unlike these, our work achieves this goal without taking the complexity of probabilistic modeling. Another parallel line of work pursues disentanglement of object properties by adopting a spatial transformer decoder – a special decoder designed to accept explicit values of object size and position (Jaderberg et al., 2015; Eslami et al., 2016; Crawford & Pineau, 2019b; Lin et al., 2020b; Jiang & Ahn, 2020; Deng et al., 2020; Chen et al., 2021). However, these cannot scale to other factors of variation such as color or material. Prabhudesai et al. (2021) seek object and factor-level disentanglement but require the use of 3D viewpoint, occupancy labels, and an auxiliary consistency loss. Learning neural networks composed of independent competing modules has been explored by Goyal et al. (2021b; 2020); Lamb et al. (2021); Goyal et al. (2021a). Our prototype learning is related to van den Oord et al. (2017); Liu et al. (2021); Caron et al. (2020) and our concept memory attention is related to (Ramsauer et al., 2021). However, these works do not focus on object-centric learning or intra-slot disentanglement.

Disentangling single-vector scene representations has been pursued in Chen et al. (2016); Higgins et al. (2017); Kim & Mnih (2018); Kumar et al. (2017); Chen et al. (2018); Locatello et al. (2019); Montero et al. (2021). Parallel to these, Hu et al. (2022); Zhou et al. (2020) pursue block-level disentanglement but on 3D mesh data and in single object scenes. Stammer et al. (2021) use prototypes to learn block-wise disentangled codes. However, this is not unsupervised and does not deal with multiple objects, unlike ours. Several metrics have also been proposed alongside the above works to measure disentanglement in single vector representations (Higgins et al., 2017; Kim & Mnih,

2018; Chen et al., 2018; Eastwood & Williams, 2018; Ridgeway & Mozer, 2018; Kumar et al., 2018; Andreas, 2019). For slot representations, several metrics have been proposed but these do not apply to block-level disentanglement within slots (Racah & Chandar, 2020; Xie et al., 2022; Nanbo et al., 2020). The metric of Dang-Nhu (2022) applies to block-level disentanglement. However, it involves simultaneously optimizing slot permutation and factor prediction which could be unstable. Also, it requires building a much larger feature-importance matrix which can be extremely slow.

## 5 EXPERIMENTS

**Datasets.** We evaluate our model on three datasets: CLEVR-Easy, CLEVR-Hard, and CLEVR-Tex. These are variants of the original CLEVR dataset (Johnson et al., 2017) having large object sizes to make object properties such as shape or texture clearly visible in the image. These datasets have different difficulty levels: *i)* In CLEVR-Easy, there are three ground-truth factors of variation, i.e., color, shape, and position while the material and the size are fixed. As in the original CLEVR, the color takes values from a small palette of 8 colors. *ii)* In CLEVR-Hard, all 5 properties i.e. color, shape, position, material, and size are the factors of variation. Furthermore, the color takes values from a much larger palette containing 137 colors. *iii)* CLEVR-Tex is derived from the original CLEVR-Tex dataset (Karazija et al., 2021a). Here, the objects and the backgrounds can take textured materials from a total of 57 possible materials and is much more complex than the previous two datasets. In addition to the material, the object shapes and positions are also varied. Our model only sees the raw images as input and no other supervision is provided. For evaluation, we use ground-truth object masks and factor labels. See Fig. 5 for sample images of these datasets. We release our code and the datasets here[1].

**Baselines.** We compare our model with three state-of-the-art unsupervised object-centric representation models: IODINE (Greff et al., 2019), Slot Attention (Locatello et al., 2020), and SLATE (Singh et al., 2022a). IODINE represents the class of VAE-based object-centric models while Slot Attention and SLATE are two deterministic object-centric models. The number of factor representations per slot for the baseline models is their slot size since every dimension acts as a factor representation. Following their original papers, we set this to 64. In our model, the number of factor representations per slot is the number of blocks $M$. We use 8, 16, and 8 blocks for CLEVR-Easy, CLEVR-Hard, and CLEVR-Tex, respectively. Importantly, having such few factor representations, e.g., 8 or 16, is a desirable characteristic of our model because it leads to better interpretability within a slot. As such, we also test how IODINE would perform with such few factor representations. We test this by training IODINE by setting its slot size equal to the number of blocks $M$ of our model (termed as IODINE-$M$). Because using such small slot sizes deviates from IODINE's original hyperparameters, we also analyze a wide range of other slot sizes for IODINE in Fig. 6 and find that it does not lead to much improvement in IODINE's performance.

### 5.1 OBJECT AND FACTOR DISENTANGLEMENT

To compare how well the models decompose the image into slots and then the slots into the factors of object variation, we use four metrics. To evaluate object segmentation, we adopt the Adjusted Rand Index applied to the foreground objects (FG-ARI), as done in prior works (Greff et al., 2019; Locatello et al., 2020). FG-ARI has also been shown to correlate with downstream task performance (Dittadi et al., 2022). To evaluate intra-slot disentanglement, we use DCI introduced in Section 3.

**Quantitative Results.** We show a quantitative evaluation of object-level and factor-level disentanglement in Fig. 2. We note that our model significantly outperforms the baselines in all 3 datasets, nearly doubling the disentanglement and completeness-scores in CLEVR-Easy and CLEVR-Hard. In CLEVR-Tex, the other models except for SLATE completely fail in handling the visual complexity of the textured scenes while our model remains robust. To the best of our knowledge, it is for the first time in this line of research that factor-level disentanglement has been achieved on this level of textural complexity. The informativeness-score and FG-ARI also suggest that our model is superior in factor prediction and object segmentation to the other methods. In Fig. 3, we visualize the feature importance matrices used for computing the DCI scores in CLEVR-Easy. We note that our importance matrix is more sparse than other methods. We also note that deterministic baselines like SLATE and

---

[1]`https://sites.google.com/view/neural-systematic-binder`

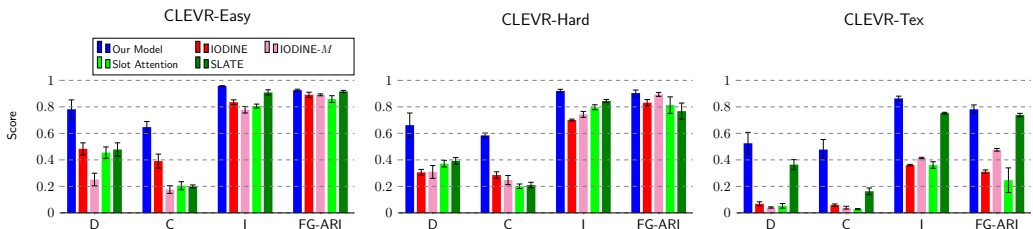

Figure 2: **Slot Learning and Intra-Slot Disentanglement.** We compare our model with the baselines in terms of Disentanglement (D), Completeness (C), Informativeness (I), and FG-ARI.

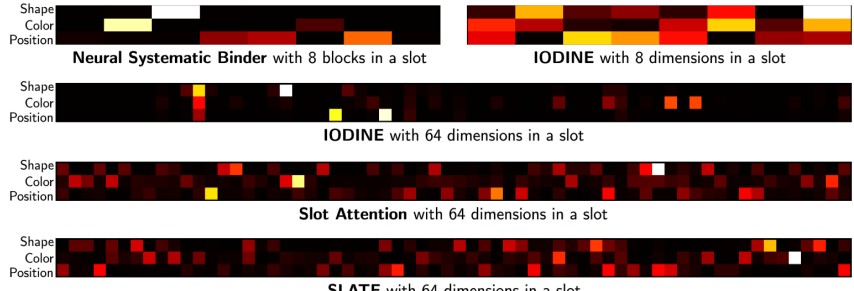

Figure 3: **Feature Importances in CLEVR-Easy.** We visualize the intra-slot feature importance matrices of the evaluated models. Each row corresponds to a ground-truth intra-object factor and each column corresponds to a factor representation. In our model, a factor representation is a block while in the baselines, a factor representation is a single dimension of the slot.

Slot Attention have significantly more active dimensions than the IODINE baseline due to the effect of the VAE prior in IODINE. This observation also correlates with lower completeness-scores of Slot Attention and SLATE relative to IODINE. Such effect of the VAE prior was also noted in Dang-Nhu (2022). Surprisingly, unlike the conventional methods, in SysBinder, such disentanglement emerges with a simple deterministic model and without applying a VAE framework.

**Qualitative Results.** In Fig. 4 and 10, we visualize the learned representation space of each block. For this, we feed a large batch of images to SysBinder, collect the $m$-th blocks, and then apply $k$-means clustering on the collected block representations. By visualizing the objects belonging to various clusters in each $m$-th block, we observe the emergence of semantically meaningful and abstract factor concepts. For CLEVR-Easy as an example, in $m = 3$ we can see shape concepts like sphere or cylinder; in $m = 2$ color concepts like cyan and blue; and in $m = 5$ the position concepts. It is remarkable that this emergence also occurs in the visually complex CLEVR-Tex dataset—for the first time in this line of research. It is remarkable to see all the spheres of CLEVR-Tex dataset are clustered together despite the fact that their low-level surface appearances and textures are very different from each other.

## 5.2 Factor-level Systematicity in Scene Composition

Since SysBinder provides knowledge blocks at the level of intra-object factors, we test whether these blocks can be composed to generate novel scenes with systematically novel factor combinations. We do this by first representing a given image as Block-Slot Representation using SysBinder. We then choose two slots, select the blocks that capture a specific object factor e.g., color, and swap the selected blocks across the two slots. The manipulated slots are then provided to the decoder to generate the scene. In Fig. 5, we show that our model can effectively generate novel images in this way. This shows that our blocks are indeed modular, allowing us to manipulate a specific factor without affecting the other properties. We also found the blocks to be interpretable and compact which enabled us to easily identify which blocks to swap in order to swap a specific factor. Furthermore, our model demonstrates this ability not only in simple scenes but also in complex textured scenes of CLEVR-Tex.

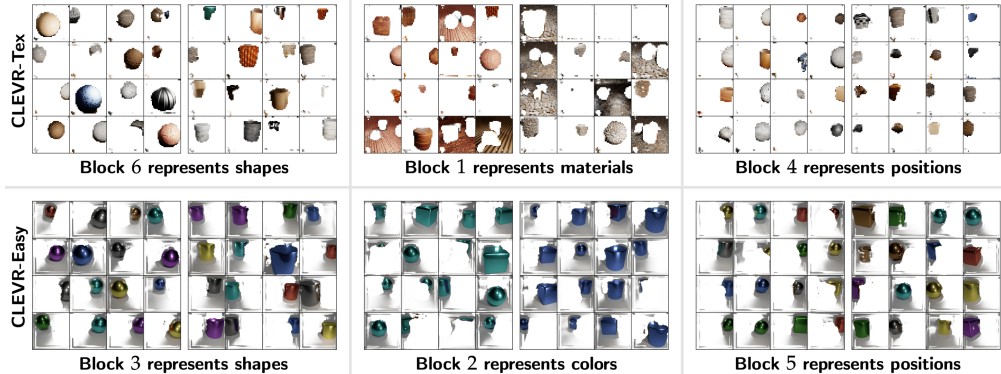

Figure 4: **Visualization of object clusters obtained by applying $k$-means on specific blocks in CLEVR-Tex and CLEVR-Easy.** Each block learns to specialize to a specific object factor e.g., shape or material, abstracting away the remaining properties. See Fig. 10 for clusters of all datasets.

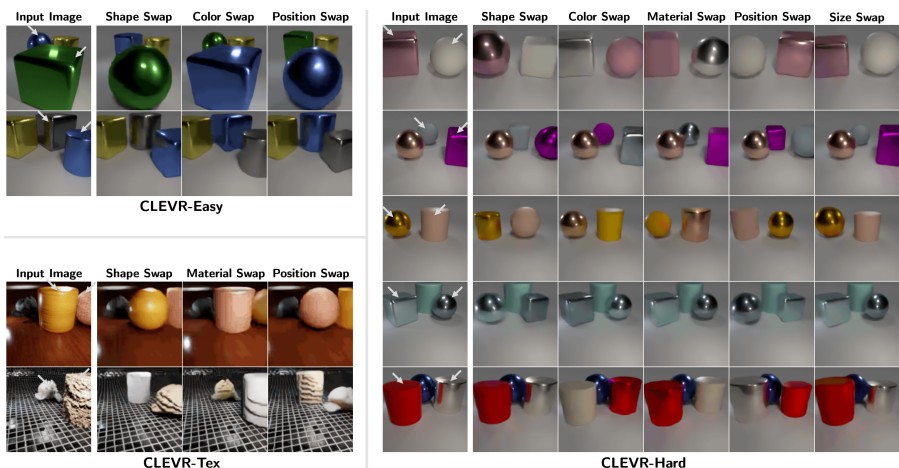

Figure 5: **Visualization of Factor-level Scene Manipulation.** We manipulate a given scene by swapping a specific factor of two objects in the scene. For a given scene, we choose two objects and swap their shape, color, position, material, and size. White arrows shown on the input images point to two objects in each scene whose properties are swapped. See also, Figures 11, 12, and 13.

## 5.3 ANALYSIS

In this section, we analyze several model variants and hyperparameters. We perform these evaluations on the CLEVR-Easy dataset and report the results in Fig. 6.

**Role of Architectural Components of SysBinder.** In Fig. 6 (a), we analyze the role of various architectural components of our model. First, we compare the effect of sharing the parameters of the RNN used during block update (i.e. the GRU and the MLP) relative to having separate RNN parameters for each block. While we do not find a significant difference in performance, nevertheless, having a separate RNN per block may serve as a more flexible and modular design for more complex large-scale problems. Second, we test the performance by removing the RNN used during the block refinement. We find that this has a significant adverse effect on the segmentation quality as well as the DCI performance, showing that the RNN is a crucial component. Third, we test the role of having the concept memory bottleneck by removing it. We find that while this does not affect the segmentation and informativeness of the slots, however, it significantly reduces the disentanglement and the completeness-scores, showing the importance of block bottleneck in enabling specialization of blocks to various intra-object factors. Fourth, we test the role of the block coupling used as the interface between SysBinder and the transformer decoder. We test this by training a variant of our model in which we provide all the blocks as a single set to the decoder. We find that performance

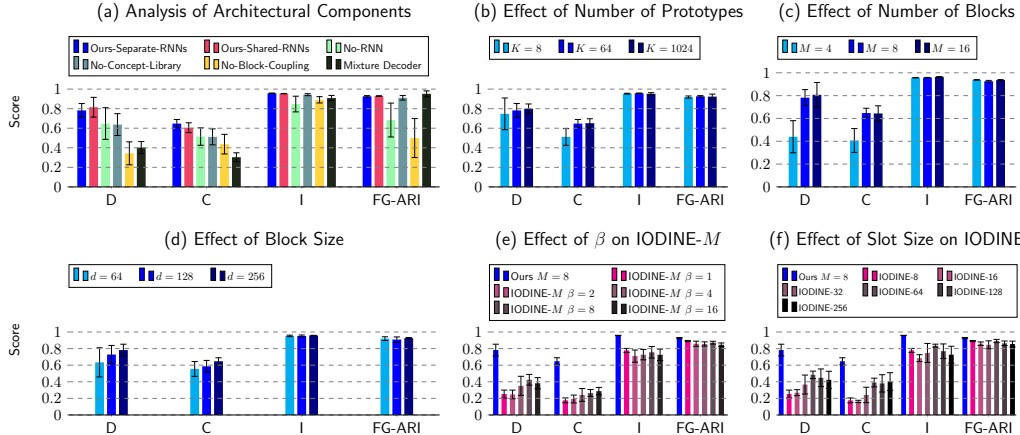

Figure 6: **Analysis.** We evaluate the role of various design choices and hyperparameters for our model and the baseline IODINE. In all plots, we report the DCI and the FG-ARI.

drops severely with some seeds failing completely. This is because without knowing the role of each block and their membership information, the decoder is unable to reconstruct well.

**Role of Transformer Decoder.** In Fig. 6 (a), we also analyze the importance of the transformer decoder in our model and whether using the mixture-based decoder, a common choice in several object-centric learning methods, can also provide a similar performance or not. To implement this variant, we adopt the decoder design of Locatello et al. (2020). We decode each object component by concatenating the blocks of a slot and providing it to a spatial broadcast decoder Watters et al. (2019). Interestingly, we find that this leads to a large drop in the disentanglement and the completeness scores, showing the important role of our expressive decoder.

**Effect of Number of Prototypes, Number of Blocks, and Block Size.** In Fig. 6 (b), (c), and (d), we study the effect of the number of prototypes $K$, the number of blocks $M$, and block size $d$ on the DCI performance. First, with the increasing number of prototypes, we note improvement, especially in the completeness-score. This suggests that more prototypes increase the expressiveness of each block and thus fewer blocks are needed to represent a single factor. Second, for the number of blocks $M = 4$, we see low disentanglement and completeness scores. Because this dataset has 3 factors of object variation, this suggests that 4 blocks may be too few for them to train well. However, going to $M = 8$ and $M = 16$, the performance improves and then saturates. Third, with increasing block sizes, we note a steady increase in performance.

**Analysis of IODINE.** In Fig. 6 (e) and (f), we analyze IODINE. We test the effect of adopting the $\beta$-VAE objective (Higgins et al., 2017) and the effect of varying the slot size. While larger $\beta$ leads to a slight increase in the disentanglement-score and the completeness-score, it still remains significantly worse than ours. Large $\beta$ also leads to a slight drop in the informativeness-score due to the greater regularizing effect of the KL term. Increasing the slot size $d$ which is equivalent to increasing the number of factor representations in IODINE slightly improves performance but still remains worse than our method. We also note that while we test IODINE with a slot size as large as 256, having few factor representations, e.g., 8 or 16, provides better interpretability within a slot.

## 6 CONCLUSION

We propose a new structured representation, Block-Slot Representation, and its new binding mechanism, Neural Systematic Binder (SysBinder). Unlike traditional Slot Attention, SysBinder learns to discover the factors of variation within a slot. Also, SysBinder provides the factor representations in a vector form, called block, rather than a single-dimensional variable. Using SysBinder, we also introduce a new unsupervised object-centric learning method for images. In achieving this, we also note the importance of autoregressive transformer-based decoding as well as our novel block-coupling approach. Our results show various benefits in disentanglement produced by the proposed model.

## REPRODUCIBILITY STATEMENT

We release all the resources used in this work, including the code and the datasets, at the project link: `https://sites.google.com/view/neural-systematic-binder`.

## ETHICS STATEMENT

Future extensions and applications arising from our work should be mindful of the environmental impact of training large-scale models. They should actively avoid its potential misuse in surveillance or by generating synthetic images with malicious intent. However, it is unlikely that the model in its current form would lead to such an impact in the near future. Our model also has the potential for making a positive impact in areas such as scene understanding, robotics, and autonomous navigation.

## ACKNOWLEDGEMENT

This work is supported by Brain Pool Plus (BP+) Program (No. 2021H1D3A2A03103645) and Young Researcher Program (No. 2022R1C1C1009443) through the National Research Foundation of Korea (NRF) funded by the Ministry of Science and ICT. The authors would like to thank the members of Agent Machine Learning Lab for their helpful comments.

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

# A   PSEUDO-CODE

---

**Algorithm 1 Neural Systematic Binder.** The algorithm receives the set of input features $\mathbf{E} \in \mathbb{R}^{L \times D}$; the number of slots $N$; the number of blocks $M$; and the block size $d$. The model parameters include: the linear projections $k$, $q$, $v$ with output dimension $Md$; GRU and MLP networks for each $m$ i.e. $\text{GRU}_{\phi_1} \dots \text{GRU}_{\phi_M}$ and $\text{MLP}_{\phi_1} \dots \text{MLP}_{\phi_M}$; the learned concept memories $\mathbf{C}_1, \dots, \mathbf{C}_M \in \mathbb{R}^{K \times d}$; and a Gaussian distribution's mean and diagonal covariance $\boldsymbol{\mu}, \boldsymbol{\sigma} \in \mathbb{R}^{Md}$.

---

```
01:  S = Tensor(N, Md)
02:  S ~ N(μ, σ)
03:  E = LayerNorm(E)
04:  for t = 1...T sequentially:
05:         S = LayerNorm(S)
06:         A = Softmax(1/√Md q(S) · k(E)ᵀ, axis='slots')
07:         A = A / A.Sum(axis='inputs', keepdim=True)
08:         U = A · v(E)
09:         for n = 1...N and m = 1...M in parallel:
10:                s_{n,m} = GRU_{φ_m}(state=s_{n,m}, update=u_{n,m})
11:                s_{n,m} += MLP_{φ_m}(LayerNorm(s_{n,m}))
12:                s_{n,m} = Softmax(1/√d s_{n,m} · C_mᵀ, axis='prototypes') · C_m
13:  return S
```

---

# B   ADDITIONAL RELATED WORK

**Unsupervised Object-Centric Learning.** Object-centric learning has also been explored extensively in dynamic scenes (Kosiorek et al., 2018; Stanić & Schmidhuber, 2019; Jiang et al., 2020; Crawford & Pineau, 2019a; Lin et al., 2020a; Wu et al., 2021; Singh et al., 2021; He et al., 2019; Greff et al., 2017; Van Steenkiste et al., 2018; Veerapaneni et al., 2019; Watters et al., 2019; Weis et al., 2020; Du et al., 2020; Kipf et al., 2021; Kabra et al., 2021; Zoran et al., 2021; Besbinar & Frossard, 2021; Creswell et al., 2020; 2021; Elsayed et al., 2022); and in 3D scenes (Chen et al., 2021; Henderson & Lampert, 2020; Crawford & Pineau, 2020; Stelzner et al., 2021; Du et al., 2021b; Kabra et al., 2021; Sajjadi et al., 2022; Nanbo et al., 2020; Singh et al., 2022b). However, adopting Block-Slot Representation in dynamic and 3D scenes is a future direction and is orthogonal to our current focus. Aside from these auto-encoding-based methods, alternative frameworks have also shown promise, notably compositional energy-based scene modeling (Du et al., 2021a; Yu et al., 2021), complex-valued neural networks Lowe et al. (2022), and reconstruction-free representation learning (Caron et al., 2021; Löwe et al., 2020; Wang et al., 2022; H'enaff et al., 2022; Wen et al., 2022; Baldassarre & Azizpour, 2022). However, these methods do not disentangle object properties.

# C   ADDITIONAL RESULTS

## C.1   EXTENDED ANALYSIS OF OUR MODEL

In this section, we perform additional ablations of our model that shed further light on it.

In Fig. 7, we show all the ablation results together, including those that we could not include in the main paper due to a limitation of space. Below, we analyze these ablation results.

**Sharing the concept-memory across blocks.** We test what would be the effect of sharing not just the RNNs across the blocks but also sharing the concept memory across the blocks. We can see that as the level of weight-sharing increases, the 'completeness' score decreases. In other words, with more weight-sharing among blocks, the model tends to use more blocks on average to represent a single factor. However, as this is not a drastic degradation, we can say that weight-separation is not the major factor responsible for the emerged disentanglement. What is crucial for getting good disentanglement is doing each block's prototype attention and refinement independently.

**Concatenating blocks to construct a slot instead of Block-Coupling.** In this ablation, we test an alternative to the block-coupling approach: *what if we simply concatenate the blocks in a slot and directly provide the set of slots as input to the transformer decoder?* We find that this variant

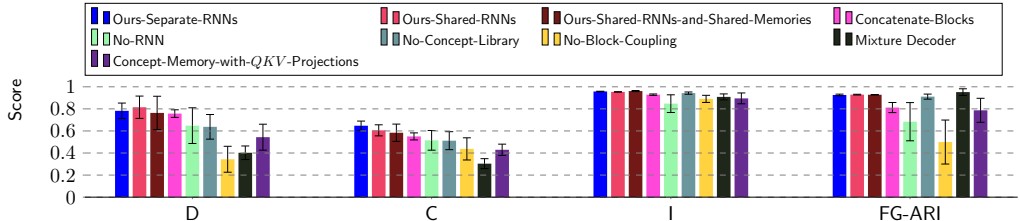

Figure 7: **Extended Analysis of Our Model.** We report Disentanglement (D), Completeness (C), Informativeness (I), and FG-ARI.

achieves a significantly worse performance in the FG-ARI score while also achieving a slightly worse performance in the DCI scores.

**Linear projections in concept-memory attention.** In this ablation, we test the effect of applying linear projections on the query, key, and value when performing attention on the concept memory as follows:

$$\mathbf{s}_{n,m} = \left[ \underset{K}{\mathrm{softmax}} \left( \frac{q_m(\mathbf{s}_{n,m}) \cdot k_m(\mathbf{C}_m)^T}{\sqrt{d}} \right) \right] \cdot v_m(\mathbf{C}_m).$$

Here, $q_m$, $k_m$, and $v_m$ are the per-block linear projections for computing the queries, the keys, and the values. We find that this variant of our model is much more unstable and performs significantly worse than our default architecture.

## C.2  ANALYSIS OF CONCEPT MEMORY AND PROTOTYPES

In this section, we qualitatively analyze several questions regarding the concept-memory bottleneck and the prototypes.

*Does concept memory attention perform hard selection of a single prototype?* We analyze the attention weights of a block on the concept memory in Fig. 8. We note that the weights are not one-hot and thus the model is not making a single hard selection of prototype. Rather, it is performing a soft selection of multiple activated prototypes. Thus, its capacity is not limited to the $K$ prototype vectors but can be much more flexible.

*How is the coverage of prototype usage?* We analyze the average activation of each prototype across a large batch of input images in the CLEVR-Easy dataset. We show these average activations in Fig. 9. We find that a large majority of prototypes seem to be active.

## C.3  BLOCK-WISE OBJECT CLUSTERS

## C.4  COMPUTATIONAL REQUIREMENTS

In Table 1, we compare the memory requirements of our model with respect to IODINE, our main baseline that pursues intra-slot disentanglement. Here, we can see that our memory consumption is better than that of IODINE. We also note that our training speed is slightly slower than that of IODINE. However, we must interpret these results with care and take into account the fact that IODINE performs significantly worse than our model when it comes to disentanglement performance.

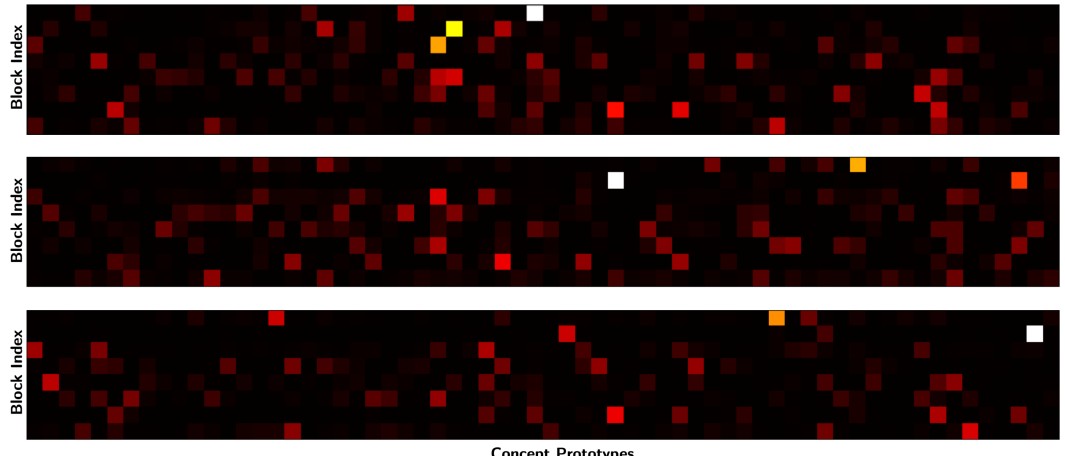

Figure 8: **Prototype Attention Weights.** Three instances of blocks attending to the prototypes in the concept memory. A cell $(m, k)$ in the visualized matrices corresponds to a block $m = 1, \ldots, 8$ and a prototype vector $k = 1, \ldots, 64$ from the concept memory of the corresponding block. We note that the attention patterns are not one-hot selection of a single prototype. Instead, it is a soft and flexible linear combination of multiple prototypes.

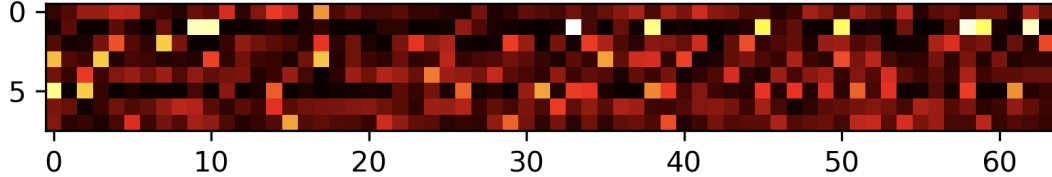

Figure 9: **Prototype Coverage.** We show the attention weight of each prototype averaged over a batch of 300 input images of CLEVR-Easy dataset. Each cell $(m, k)$ in the visualized matrix corresponds to a block $m = 1, \ldots, 8$ and a prototype $k = 1, \ldots, 64$ from its corresponding concept memory $\mathbf{C}_m$.

|  | IODINE | Our Model |
|---|---|---|
| Memory Consumption | 37.30 GiB | **31.01 GiB** |
| Time per Training Iteration | **0.84s** | 1.01s |

Table 1: **Comparison of computational requirements between IODINE and our model.** For fair comparison, all values are measured on training runs with the same batch size of 40.

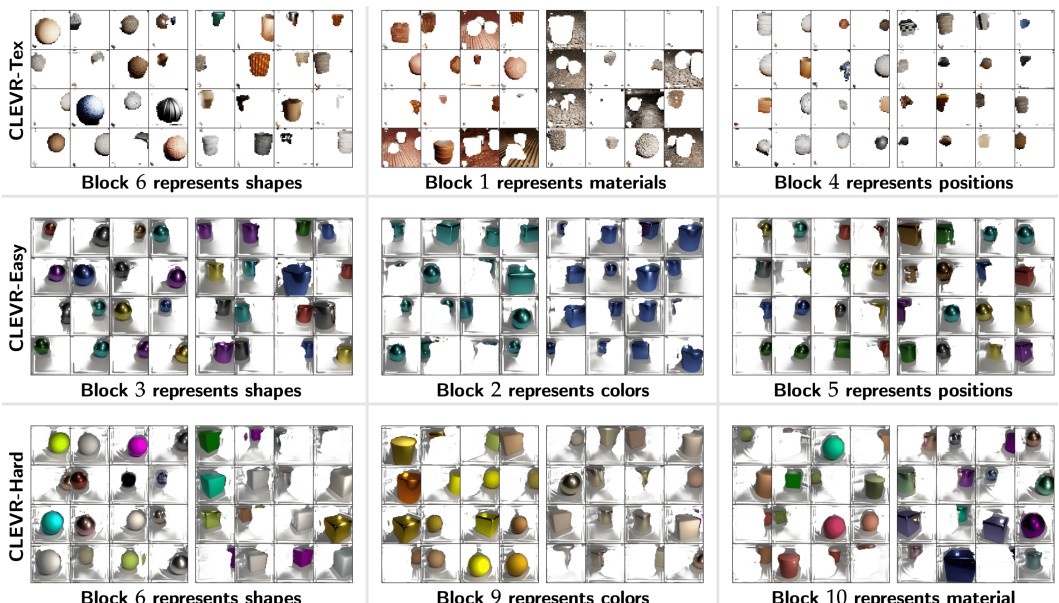

Figure 10: **Visualization of object clusters obtained by applying $k$-means on specific blocks in CLEVR-Easy, CLEVR-Hard, and CLEVR-Tex.** Each block learns to specialize to a specific object factor e.g., shape or color, abstracting away the remaining properties.

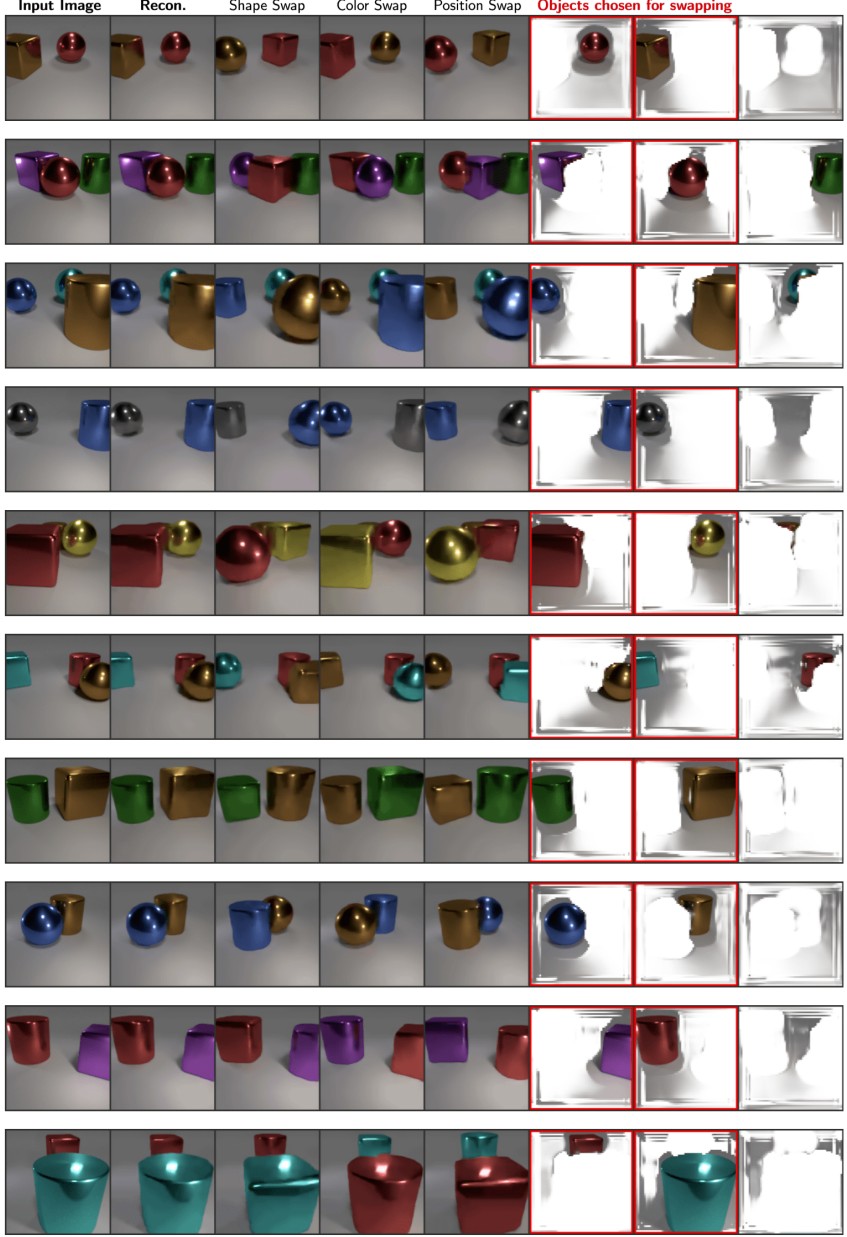

Figure 11: Additional samples of factor-swapping in CLEVR-Easy.

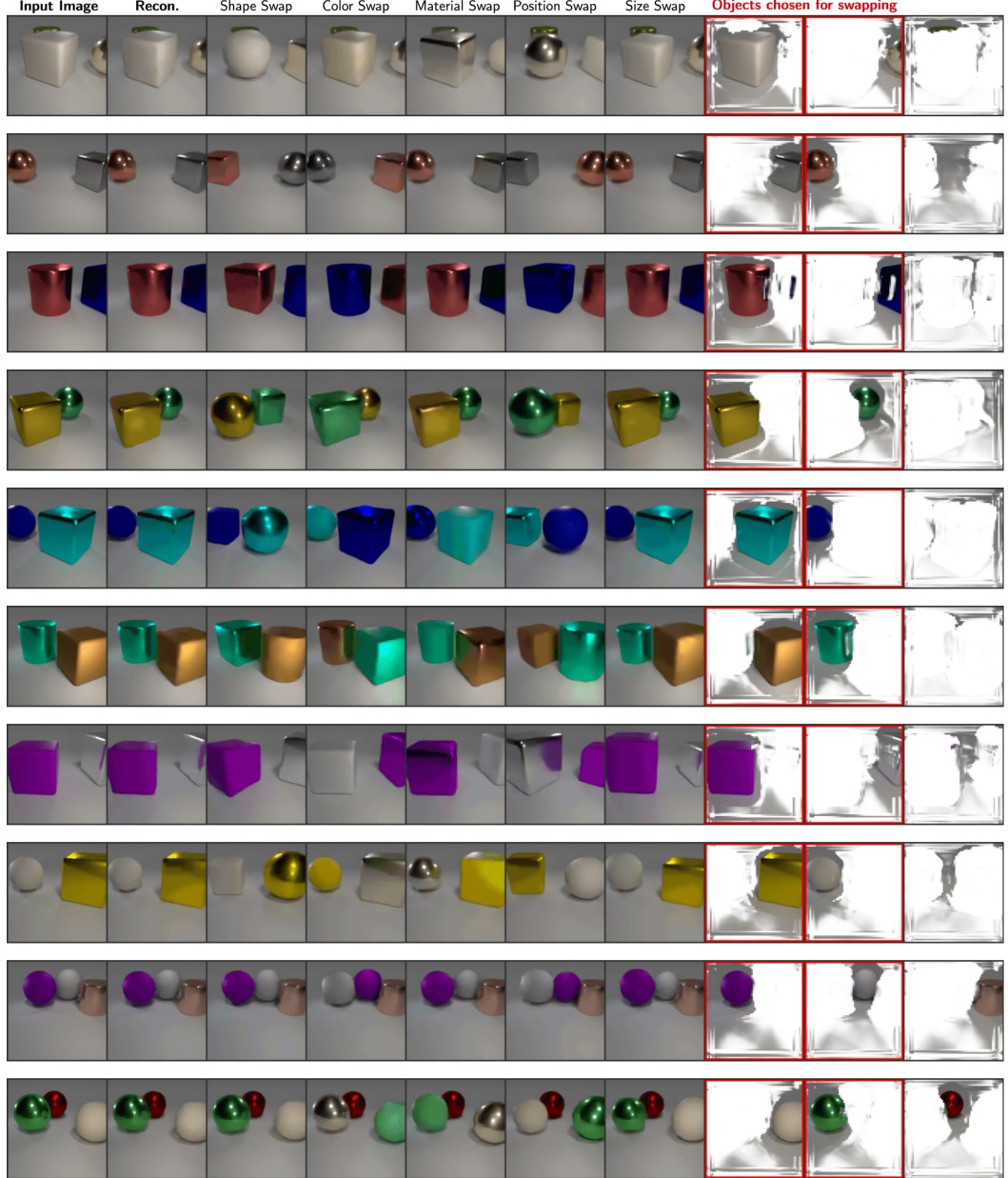

Figure 12: Additional samples of factor-swapping in CLEVR-Hard.

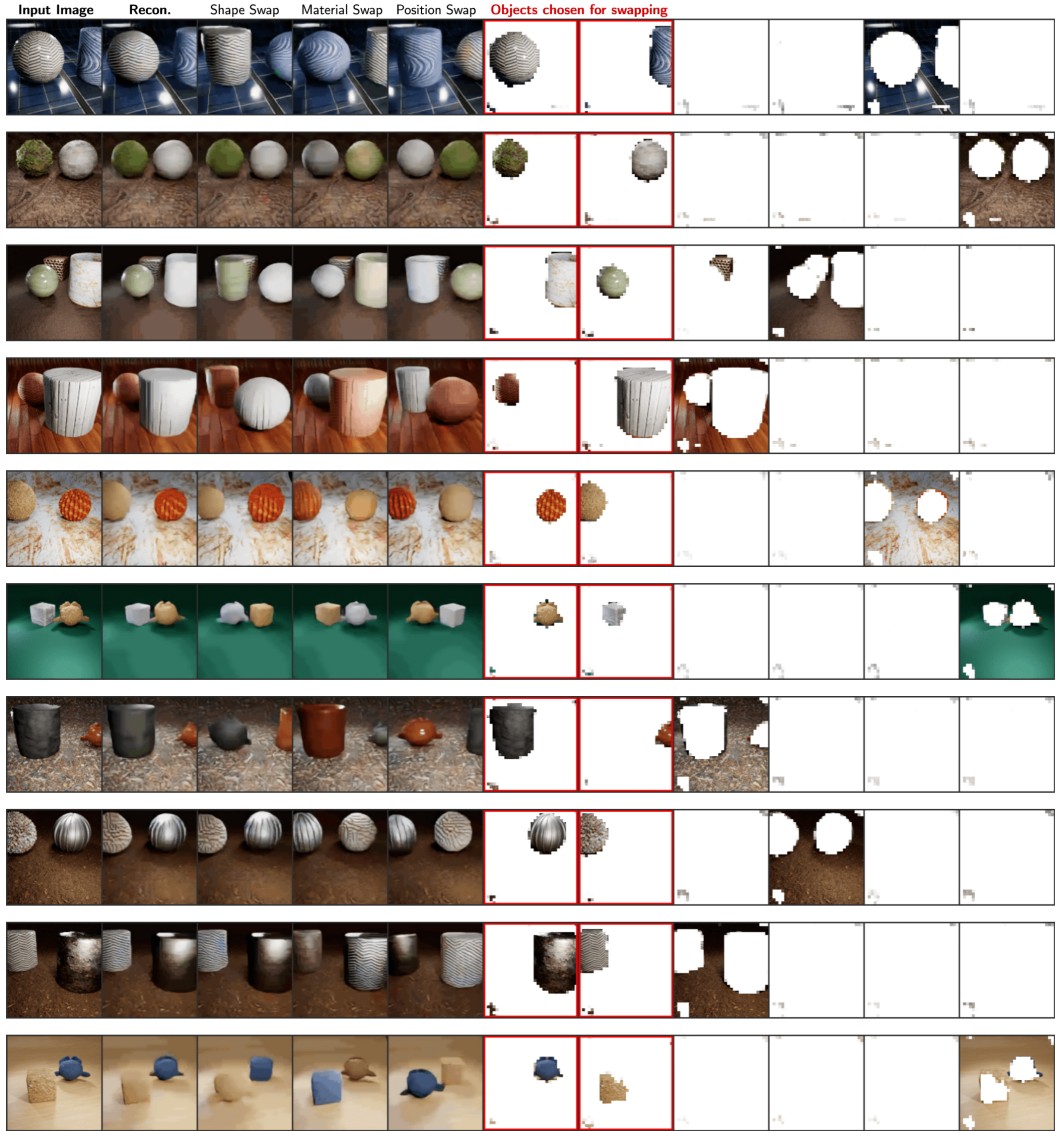

Figure 13: Additional samples of factor-swapping in CLEVR-Tex.

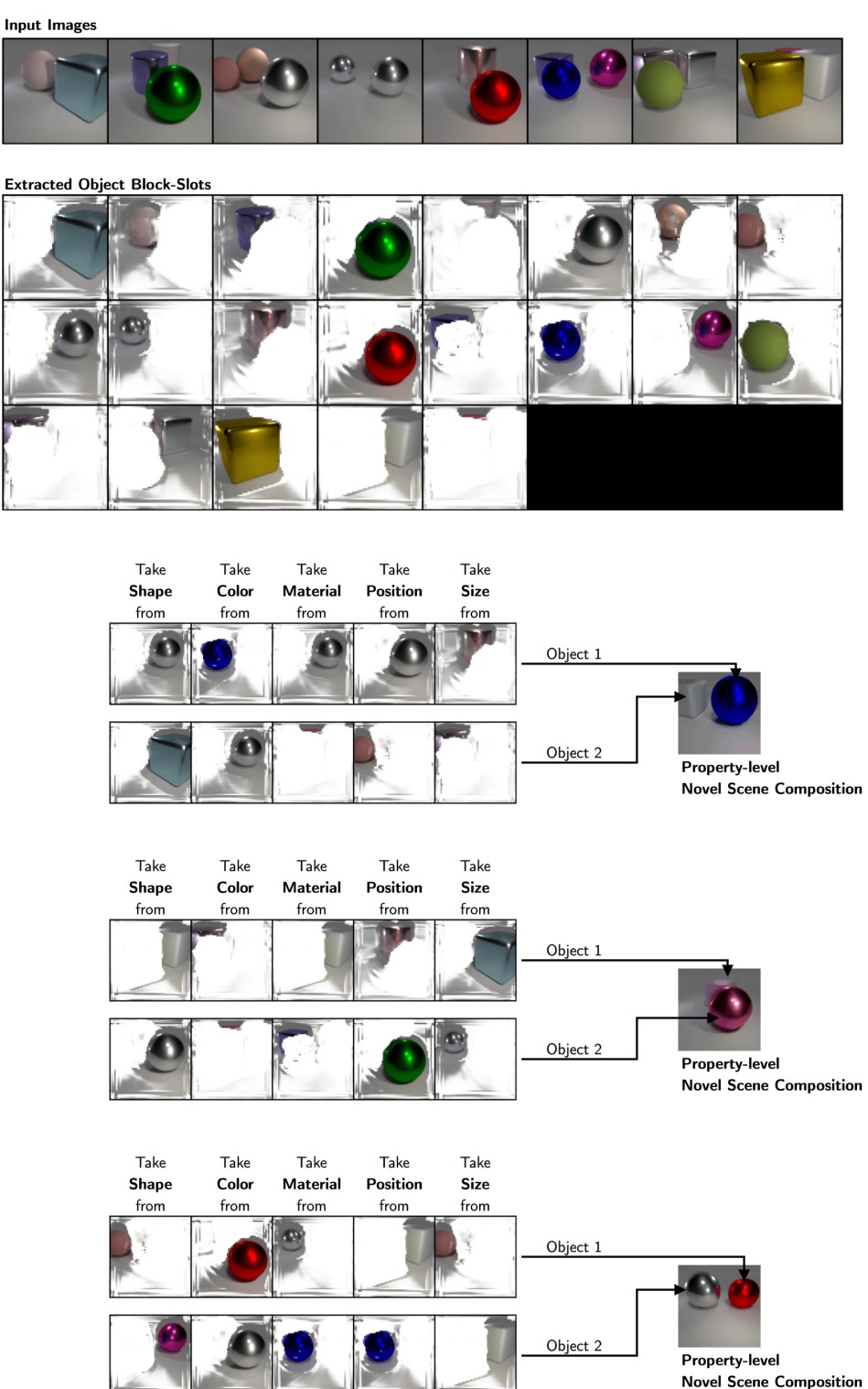

Figure 14: **Systematic Scene Generation in CLEVR-Hard.** We are given 8 input images from which we extract slots. Using these extracted slots, we compose new objects by combining object properties into novel combinations. By decoding these composed novel slots, we show that we can generate novel scenes.

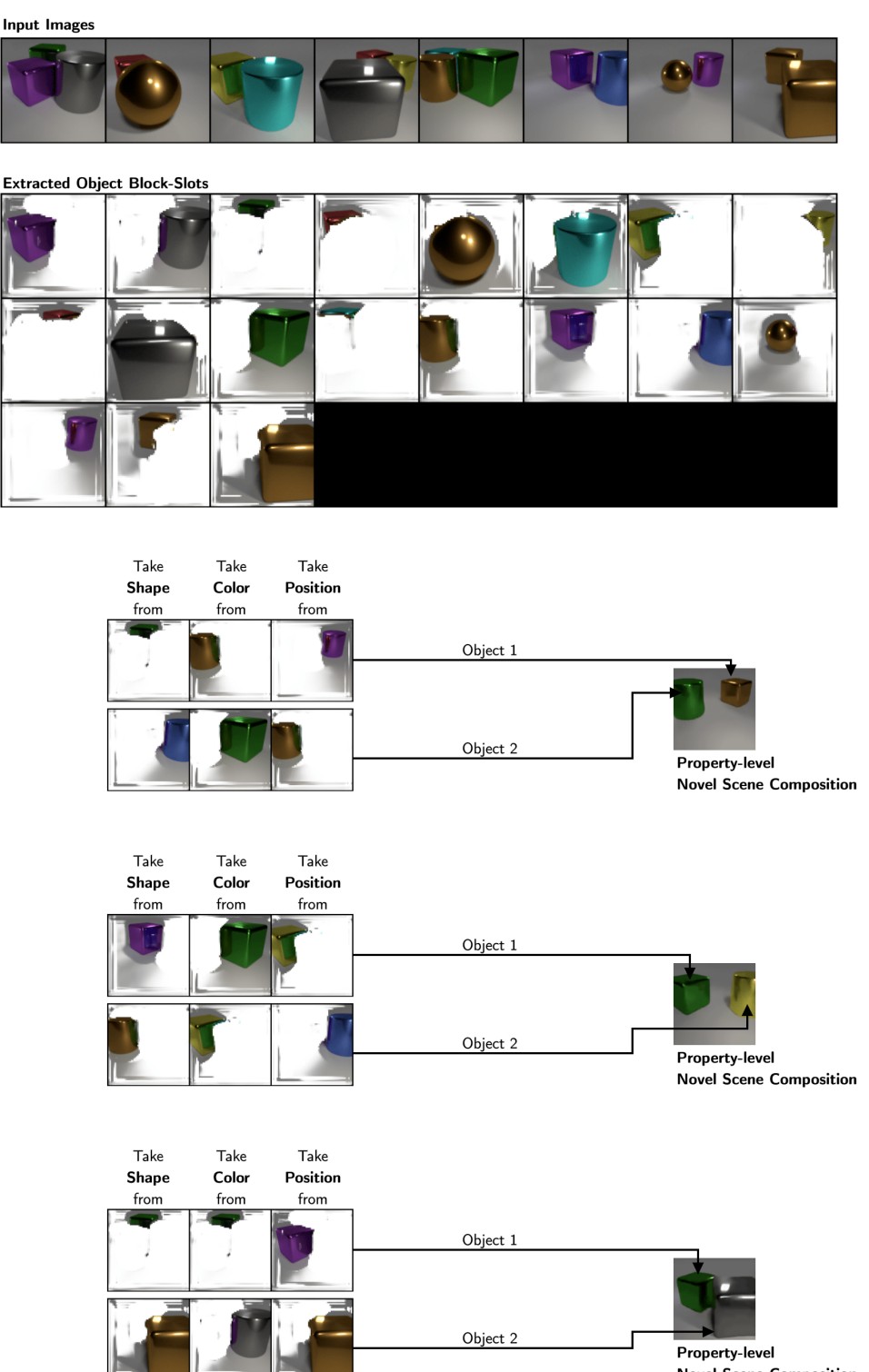

Figure 15: **Systematic Scene Generation in CLEVR-Easy.** We are given 8 input images from which we extract slots. Using these extracted slots, we compose new objects by combining object properties into novel combinations. By decoding these composed novel slots, we show that we can generate novel scenes.

# D  ADDITIONAL IMPLEMENTATION DETAILS

| Module | Hyperparameter | Dataset | | |
|--------|----------------|---------|---|---|
| | | CLEVR-Easy | CLEVR-Hard | CLEVR-Tex |
| General | Batch Size | 40 | 40 | 40 |
| | Training Steps | 200K | 200K | 400K |
| SysBinder | Block Size | 256 | 128 | 256 |
| | # Blocks | 8 | 16 | 8 |
| | # Prototypes | 64 | 64 | 64 |
| | # Iterations | 3 | 3 | 3 |
| | # Slots | 4 | 4 | 6 |
| | Learning Rate | 0.0001 | 0.0001 | 0.0001 |
| Transformer Decoder | # Decoder Blocks | 8 | 8 | 8 |
| | # Decoder Heads | 4 | 4 | 8 |
| | Hidden Size | 192 | 192 | 192 |
| | Dropout | 0.1 | 0.1 | 0.1 |
| | Learning Rate | 0.0003 | 0.0003 | 0.0003 |
| dVAE | Learning Rate | 0.0003 | 0.0003 | 0.0003 |
| | Patch Size | $4 \times 4$ pixels | $4 \times 4$ pixels | $4 \times 4$ pixels |
| | Vocabulary Size | 4096 | 4096 | 4096 |
| | Temperature Start | 1.0 | 1.0 | 1.0 |
| | Temperature End | 0.1 | 0.1 | 0.1 |
| | Temperature Decay Steps | 30000 | 30000 | 30000 |

Table 2: Hyperparameters of our model used in our experiments.

## D.1  BACKBONE IMAGE ENCODER

The backbone image encoder consists of a CNN encoder that encodes the image into a feature map. The specifications of this CNN encoder are described in Table 3. For CLEVR-Tex dataset, we found it useful to use a deeper CNN network described in Table 4. After applying the CNN, positional encodings are added to the feature map. Layer-normalization is applied on the resulting feature map followed by a 2-layer MLP with hidden dimension 192. The resulting feature map is flattened along the spatial dimensions to produce a set of input features that are then provided to the SysBinder module. The positional encodings are learned in a similar way as in Slot Attention by mapping a 4-dimensional grid of position coordinates to a 192-dimensional vector via a linear projection.

| Layer | Kernel Size | Stride | Padding | Channels | Activation |
|-------|-------------|--------|---------|----------|------------|
| Conv | $5 \times 5$ | 2 | 2 | 512 | ReLU |
| Conv | $5 \times 5$ | 1 | 2 | 512 | ReLU |
| Conv | $5 \times 5$ | 1 | 2 | 512 | ReLU |
| Conv | $5 \times 5$ | 1 | 2 | 192 | None |

Table 3: Specifications of CNN layers of the backbone image encoder in CLEVR-Easy and CLEVR-Hard datasets.

## D.2  RECURRENT BLOCK UPDATE

The block update is performed using a GRU network and a 2-layer MLP network with dimensions equal to the block size. For maintaining the separation of GRU and MLP model parameters between blocks and to apply GRU and MLP on all the $M$ blocks in parallel, we adopt the implementation of Goyal et al. (2021b).

| Layer | Kernel Size | Stride | Padding | Channels | Activation |
|-------|-------------|--------|---------|----------|------------|
| Conv | $5 \times 5$ | 2 | 2 | 512 | ReLU |
| Conv | $5 \times 5$ | 1 | 2 | 512 | ReLU |
| Conv | $5 \times 5$ | 2 | 2 | 512 | ReLU |
| Conv | $5 \times 5$ | 1 | 2 | 512 | ReLU |
| Conv | $5 \times 5$ | 1 | 2 | 192 | None |

Table 4: Specifications of CNN layers of the backbone image encoder in CLEVR-Tex dataset.

### D.3 Concept Memory

To facilitate the training of the concept memories $\mathbf{C}_1, \ldots, \mathbf{C}_M$, we parameterize prototypes as an MLP projection of a learned vector: $\mathbf{C}_{m,k} = \texttt{MLP}^{\text{prototype}}(\boldsymbol{\epsilon}_{m,k})$, where each $\boldsymbol{\epsilon}_{m,k}$ is a learned vector of dimension equal to the block size. The MLP has 4 layers with hidden dimensions 4 times the block size and the output dimension set to be equal to the block size.

### D.4 Image Tokenizer and Transformer

We use discrete VAE, or simply dVAE, as our image tokenizer. We adopt the same architecture as that used in SLATE (Singh et al., 2022a). Like SLATE, the temperature of Gumbel-Softmax (Jang et al., 2016) is decayed using cosine-annealing from 1.0 to 0.1 in the first 30K training iterations. Like SLATE, the dVAE is trained simultaneously with the other parts of the model. We use a patch size of $4 \times 4$, meaning that a $128 \times 128$-sized image would be represented as a sequence of 1024 discrete tokens. To learn to generate these tokens auto-regressively, we adopt the same transformer decoder and configurations as SLATE without any architectural or significant hyperparameter changes.

### D.5 Training

Similar to SLATE (Singh et al., 2022a), we split the learning rates of the dVAE image tokenizer, the SysBinder encoder, and the transformer decoder. The learning rates and the full set of hyperparameters are provided in Table 2. During training, the learning rate is linearly warmed up to its peak value in the first 30K training iterations and then decayed exponentially with a half-life of 250K iterations.

## E Additional Experimental Details

### E.1 Visualization of Object Clusters using $K$-means on Blocks

In this section, we provide a more detailed description of the visualizations of Fig. 10. To generate these visualizations, we first take a large collection of images. On these, we apply BSA and extract $\mathcal{S} = \{(\mathbf{s}_i, \mathbf{x}_i^{\text{attn}})\}$: a collection of slots $\mathbf{s}_i$ paired with its corresponding object image $\mathbf{x}_i^{\text{attn}}$ of the obtained by multiplying the input image with the attention mask of the slot $\mathbf{s}_i$. From this collection, we take $m$-th blocks as $\mathcal{S}_m = \{(\mathbf{s}_{i,m}, \mathbf{x}_i^{\text{attn}})\}$ for each block index $m$. We then apply $K$-means on the block vectors in $\mathcal{S}_m$ to obtain clusters of blocks. We use $K = 12$. In Fig. 10, we visualize the blocks inside some of the clusters by showing the object that those blocks are representing. In the top-left of Fig. 10, we show two of the clusters for the block index 6 for CLEVR-Tex by showing the object images of blocks that lie in those clusters. It shows that all the sphere objects had a representation of block 6 close to each other. Similarly, all the cylinder objects also have block 6 representations that are close to each other. This shows which concept each block is capturing and which it is abstracting away. That is, in this case, block 6 is capturing the shape factor of the object but abstracting away other properties such as material or position.

### E.2 Selecting Blocks for Swapping

We selected the blocks representing a specific factor by first applying $K$-means on each $m$-th block as in the qualitative analysis of Fig. 10. We then manually inspected the resulting object clusters, and identified, for each block, which factor a block is specializing in.

## F   LIMITATIONS AND DISCUSSION

The proposed model also shows some limitations. First, although our model significantly outperforms the baselines, the absolute performance is still far from perfect. Second, the vector-formed factors in our model make slots larger and thus take more computation. Lastly, our transformer decoder can be computationally more expensive than the mixture decoders used in the conventional approaches.

A few future directions are as follows. First, it would be interesting to extend the model to make it work on more complex natural scenes even though the current dataset is already much more complex than those in a similar line of research (Greff et al., 2019; Locatello et al., 2020). Second, it would be interesting to study the multi-modal aspect of the proposed model, particularly with language. Finally, an extension to video learning is also interesting.

