# OpenReview forum: "Neural Systematic Binder"
_ICLR.cc/2023/Conference — ICLR 2023 poster_

### Official Review · Reviewer_HPVD · 2022-10-23

**Confidence:** 3
**Correctness:** 3
**Technical Novelty And Significance:** 2
**Empirical Novelty And Significance:** 2
**Recommendation:** 6

**Clarity, Quality, Novelty And Reproducibility:**

> Clarity

Good.

> Quality

Good.

> Novelty

Normal. As discussed in the related work section, there already exist some efforts along the path of within-slot disentanglement although the proposed approach seems to generalize well and does not require explicit supervision. Similar ideas also have been investigated in the Part-and-Sum detection Transformer along the path of the DETR community.

> Reproducibility

No code is available.

**Details Of Ethics Concerns:**

No ethics concerns.

**Strength And Weaknesses:**

> Strengths

✅ The presented idea is interesting and the illustration is clear.

✅ The approach section is well-organized into two subsections including (i) the details of the proposed block-slot attention and (ii) how to
perform object-centric learning with a block-slot attention scheme.


> Weaknesses

❎ According to Figure 1 and the Approach section, we can see that the proposed approach might bring significant extra computation overhead compared
to the original baseline approach. The authors should report detailed complexity comparisons such as the number of parameters and GFLOPs.

❎ The authors only verify the proposed approach on the very toy benchmarks like CLEVR-Easy, CLEVR-Hard, and CLEVR-Tex.
It would be great if the authors could extend the approach to some real-world benchmarks following very recent works such as SAVi[1] and SAVi++[2]. Besides,  the authors are encouraged to discuss more details and compare the proposed approach to GENESIS-V2[3] empirically if possible.

[1] SlotFormer: Unsupervised Visual Dynamics Simulation with Object-Centric Models

[2] SAVi++: Towards End-to-End Object-Centric Learning from Real-World Videos

[3] GENESIS-V2: Inferring Unordered Object Representations without Iterative Refinement

❎ The emergence of semantically meaningful abstract concepts in blocks shown in Figure 4 is very impressive. The authors are encouraged to conduct detailed
ablation experiments to investigate which component is the key to ensuring the emergence of semantically meaningful abstract concepts and whether this conclusion
still holds on some real-world benchmarks such as ImageNet and COCO.

**Summary Of The Paper:**

This paper presents a block-slot object-centric representation learning scheme
to use a set of sub-slot representations, aka., block representations, to explicitly encode different properties that
are conventionally encoded with only one slot representation.
The authors show the proposed disentanglement scheme performs better and enjoys better interpretability.

This idea also seems related to the Part-and-Sum detection Transformer[1] used to perform visual relationship
detection, where the authors also propose to use part queries to explicitly
encode the different partial information that is originally encoded with one sum query.
The authors are highly encouraged to discuss whether the proposed method can be applied to
broader vision tasks based on DETR such as object detection and segmentation.

[1] Visual relationship detection using part-and-sum transformers with composite queries, CVPR2021

**Summary Of The Review:**

How to perform within-slot disentanglement to real-world images in an unsupervised manner is a grand challenge. This work presents some encouraging results in this direction. However, the experiments are relatively weak and the authors are encouraged to extend the proposed approach to more challenging benchmarks, which might be necessary to attract broader attention from the research community. I would like to increase the ratings if the authors could well address the above concerns.

---

> ### Author Response · Authors · 2022-11-15
> **Response to Reviewer HPVD**
>
> Thank you for raising these relevant points. We address them below.
>
> > Extend to real-world benchmarks
> >
>
> We respond to this in the common response above.
>
>
> > Authors should compare the computational requirements.
> >
>
> Thank you for pointing. Here, we compare the computational requirements of our model with IODINE — the state-of-the-art model that pursues intra-slot disentanglement. All numbers are reported using the same batch size of 40 for a fair comparison.
>
> Memory Consumption
>
> - IODINE: 37.30 GiB
> - Our Model: 31.01 GiB
>
> Training Speed
>
> - IODINE: 0.84s per iteration
> - Our Model: 1.01s per iteration
>
> Here, we can see that our memory consumption is better/lower than that of IODINE. We also note that our training speed is slightly slower than that of IODINE. However, we must interpret this with care and take into account the fact that IODINE performs much worse than our model when it comes to disentanglement performance. Given the significant benefits of our model over IODINE, the computational overhead of our model is not large. We have also added these results in Table 1 in the appendix in our revised version.
>
>
>
>
> > Compare with GENESIS-v2
> >
>
> We believe that GENESIS-v2 would not be a suitable baseline for our work for the following reasons:  First, the focus of the GENESIS family of models is on coherent scene generation and not on intra-slot disentanglement like our work. Second, the result of Karazija et al. (2021) has already shown that GENESIS-v2 fails on the CLEVR and CLEVR-Tex benchmarks. GENESIS-v2 was also shown to perform significantly worse than the two baselines that we already compare with: Slot Attention and IODINE.
>
> > Conduct detailed ablation to show what is responsible for the emergence of concepts.
> >
>
> We already conduct an ablation of our model in Fig. 6 (top-left). We have also added 3 additional analyses based on the comments of the reviewers in Fig. 7 in the appendix. Alternatively, the ablation results may also be viewed at this anonymous link: [https://imgur.com/a/8JGBlbb](https://imgur.com/a/8JGBlbb).
>
> Based on these results, we believe that 1) the concept memory bottleneck applied to each block and 2) the strong auto-regressive transformer-based decoding are the two key elements that lead to the emergence of abstract disentangled concepts in the blocks.
>
> Feel free to let us know any additional ablation that you feel would add further insights about the model. We would be happy to run the experiment and add the results.
>
> > No code is available.
> >
>
> We will release our code and all datasets at the link (currently anonymized): [https://github.com/blockslotattention](https://github.com/blockslotattention).
>
> > Discuss whether the proposed method can be applied to broader vision tasks based on DETR such as object detection and segmentation.
> >
>
> While our method does provide object discovery and segmentation masks in a fully-unsupervised manner, however, in this work, our focus is on disentangling the object properties within a slot (also without any supervision).
>
> Regarding the question of dealing with real images (such as those that supervised models like DETR commonly handle), we answer it in our common response.

---

### Official Review · Reviewer_NtVS · 2022-10-24

**Confidence:** 5
**Correctness:** 3
**Technical Novelty And Significance:** 4
**Empirical Novelty And Significance:** 4
**Recommendation:** 8

**Clarity, Quality, Novelty And Reproducibility:**


I mainly have concerns with the high level presentation of the contribution, mostly in the introduction:
- *“the question [...] is not a small one but actually at the crux of this challenge.”* I would encourage you to improve the phrasing of this claim
- *“is often considered to be the deep representation learning approach towards high-level cognition.”* I am not sure I understand the meaning of this sentence
- *“The state-of-the-art methods for object-centric representations are currently based on Slot Attention”* I don’t specifically disagree with this claim but I think it would be good to provide more context (on which dataset and for which metrics?)
- *“In Slot Attention, each slot finds a local area”* … *“attended local input”* could you point me to a reference showing that the attention process of Slot Attention is local (which would imply a notion of proximity)?
- *“intra-slot disentanglement has naturally been achieved in the previous probabilistic approaches”* I agree that Slot Attention is originally trained in a deterministic setting. But if this is really the main concern, why not just add a latent space regularization to the training objective in order to encourage disentanglement? It feels like this is not the main concern here
- *“make it as generic as possible by making it a simple deterministic layer like Slot Attention”* again, is this really a concern here? This sentence is followed by *"We believe that once this is achieved, it should not be too difficult to make a probabilistic extension.”*, which seems to contradict your previous statement that the probabilistic view is overly complex
- *"considering that the single-dimension representation of a factor is difficult to use as an input to a Transformer"* could you please support this claim with references and/or arguments? This is the first time I hear about this issue
- at the beginning of section 2, I find it confusing to call your slots "block-slots". I would advise to simply state that a block-slot representation is composed of slots which are composed of blocks. The term of slot is anterior to Slot Attention (for instance in the Genesis or Iodine paper) and can refer to any kind of high level unit in your latent space, so you don't need to specialise the term

Concerning design choices, the concept memory bottleneck described at the bottom of page 3 looks very similar to an attention layer, except that none of the queries, keys and values are projected. Is this on purpose, and is there a risk of capacity problem here? Is this something common in the literature or is this a design choice that is specific to your contribution?

About your structured disentanglement metric in Section 4, there are two points I would like to discuss
- your claim that the structured disentanglement metric of Dang-Nhu doesn't apply to block level disentanglement inside slot is incorrect. In Section 5.4, Dang-Nhu applies their metric to disentanglement between the mask block and the component block inside each slot of the Genesis architecture. I think it's fine that you developed your own concurrent metric, but I would encourage you to make an accurate description of related work. Dang-Nhu also performs aggregation of feature importance inside the block, but the main difference is that they jointly optimise the slot to object matching and the probes via an EM approach. This is a more general approach that doesn't require any kind of visual information for matching. However it leads to more instability.
- I am concerned that your experimental study compares disentanglement of representations with different numbers of dimensions. The entropy quantity heavily depends on the number of dimensions via the number of terms in the sum and the log base, and I am not aware of any reference studying the impact of increasing the number of dimensions, therefore I am not sure of how to normalise results for comparison. Given that the qualitative evaluation in Figure 3 is consistent with the quantitative results, I'm fine with keeping your results as is, but I would at least warn the reader about this issue.

Finally, I noticed your experimental finding  *"We also note that deterministic models like SLATE and Slot Attention have significantly more active dimensions than VAE-based IODINE due to the regularising effect of the VAE prior. This observation also correlates with lower completeness-scores of Slot Attention and SLATE relative to IODINE."*, and wanted to point out that this is consistent with the systematic evaluation of Dang-Nhu, showing that the VAE loss increases intra-slot disentanglement on a variety of architectures.

Minor:
- the letter $K$ is used twice for the size of the concept memory and the number of ground truth factors.


**Strength And Weaknesses:**

The contribution is well scoped, and obtaining good multi layer disentanglement is a very important challenge. Results in Figure 2 and 3 evidence benefits of the proposed method with a clear gap in reported metrics compared to prior work. I however have some concerns about clarity of the presentation and some experimental choices. I believe that these concerns can be addressed during the rebuttal phase.


**Summary Of The Paper:**

This paper describes a novel structured representation of scenes with several levels of hierarchy. The representation is divided into slots which provide object compositionality, and slots have several blocks of latent dimensions which are aimed to disentangle factors of variations for each object. To obtain this representation, the encoder architecture can be seen as a generalization of the well known Slot Attention component to the Block-Slot setting. The model is trained in a standard autoencoding setting, with a decoder inspired from recent work in compositional generative modeling for scenes (SLATE, STEVE), also generalized to Block-Slots via original additions to the architecture. An experimental evaluation on different variations of the CLEVR dataset clearly demonstrates the advantages of the design choices for representation compositionality, quantified as pixel-level object separation and disentanglement.

**Summary Of The Review:**

This is a novel and original contribution building on the well known Slot Attention component. The technical description is clear, but I have raised a number of concerns about the high-level presentation and phrasing of claims. I am looking forward to discussion with the authors in order to make a stronger acceptance recommendation.

---

> ### Author Response · Authors · 2022-11-15
> **Response to Reviewer NtVS (1/2)**
>
> Thank you for the positive recommendation and the deep and thorough comments!
>
> > Revise the phrasing of the following in the introduction:
> > - “… not a small one but actually at the crux of this challenge.”
> > - “… the deep representation learning approach towards high-level cognition.”
> >
>
> Thank you for pointing to these. In the revised submission, we have clarified these lines. Feel free to suggest more comments about them and we would be happy to address them further.
>
> > “State-of-the-art methods are based on Slot Attention”. Provide more context for this statement.
> >
>
> Thank you for pointing this out. We changed this sentence as follows: “Currently, one of the most popular approaches to object-centric representation learning is based on Slot Attention (SA).”
>
> > Is there a reference showing that attention process of Slot Attention is ‘local’.
> >
>
> Thank you for raising this point. Yes, we agree that the notion of locality or proximity is not inbuilt in the core design of slot attention (However in practice, such locality emerges due to the locality in the input.) We fully agree with your point and we have thus removed the word ‘local’ in our revised submission.
>
> > Why should it be a concern whether the model is deterministic or probabilistic for achieving intra-slot disentanglement?
> >
>
> **Probabilistic models** are complex and hard to use. They require designing a prior for the latent, noisy optimization and auxiliary losses such as a KL term. They also introduce additional hyperparameters that a practitioner must carefully tune e.g. $\beta$ coefficient for the KL term or standard deviation $\sigma$ for the decoder distribution (Engelcke et al. 2019, Burgess et al 2019). In contrast, **Slot Attention** (Locatello et al. 2020) offers a deterministic alternative that works well and also outperforms the previous methods with very little to no hyperparameter tuning. This has lead to its wide adoption and success in the community and this is why deterministic models are more desired than probabilistic models.
>
> However despite this remarkable success, Slot Attention does not offer intra-slot disentanglement, raising the question **is there an architecture that can provide intra-slot disentanglement with a simple deterministic framework?**  This is indeed what our model offers: a simple architecture for achieving intra-slot disentanglement despite being deterministic.
>
> Furthermore, because our model is deterministic, it challenges a conventional belief that intra-slot disentanglement needs a probabilistic framework. Our work shows a surprising result that a deterministic framework can also achieve disentanglement and actually perform much better. This is another new knowledge for sharing with the community.
>
> > “It should not be too difficult to make a probabilistic version”. This seems to contradict the fact that deterministic models are preferred.
> >
>
> No, it is not contradictory for the following reason. We did not mean to imply that a probabilistic model is a desired thing and that our model should necessarily evolve into a probabilistic one. Rather, our intended meaning is closer to: *if one wants to make a probabilistic version, it would not be too difficult* because one would only need to deal with the uncertainty modeling part without the burden of dealing with the disentanglement problem. Nevertheless, to avoid unnecessary confusion, we have decided to remove the sentence in the revision.
>
> > “Single-dimension representation of a factor is difficult to use as input to a Transformer”. Support with arguments or references.
> >
>
> Transformer in its most standard form takes a set of vectors as input. If the input is a set of real numbers, it is not obvious how to deal with this. In this case, one should come up with some kind of a rather unnatural extra mechanism to map a real number into a vector to be used in a Transformer. For example, one may copy the numbers to make a vector or project it to a vector, but these will still not produce an informative representation like ours.
>
> > No need to specialize the term *slot* to *block-slot*.
> >
>
> Thank you for raising this point. After some thought, we have come to agree that we indeed do not need to specialize the term *slot*. We have now updated the writing and replaced the term *block-slot* with *slot*.
>
> > In related the work, the description of Dang-Nhu (2021) is incorrect. The metric of Dang-Nhu (2021) applies to block-level disentanglement as well.
> >
>
> Thank you for pointing that the metric of Dang-Nhu (2021) also applies to block-level disentanglement. We have now fixed the description of Dang-Nhu (2021) as follows:
>
> *“…The metric of Dang-Nhu (2021) applies to block-level disentanglement. However, it involves jointly optimizing slot permutation and property prediction which could be unstable. Also, it requires building a much larger feature-importance matrix which can be extremely slow.”*

---

> > ### Author Response · Authors · 2022-11-15
> > **Response to Reviewer NtVS (2/2)**
> >
> > > Concerned that experiments compare disentanglement in slots of different sizes.
> > >
> >
> > This is a great question. This is indeed one of the reasons why we compare our model with a baseline termed as IODINE-$M$ for all datasets. In IODINE-$M$, we train IODINE with a slot size equal to the number of blocks $M$ in our model. This enables an apples-to-apples comparison when it comes to number of factors per slot and shows that our model still performs much better.
> >
> > To support this further, we provide two new additional experiments:
> >
> > - **Effect of $\beta$.** For IODINE-$M$, we also tested whether applying a stronger $\beta$ in IODINE’s ELBO can alleviate its poor performance. We find that while larger $\beta$ slightly helps, the improvement is not substantial enough and it still performs much worse than our model.
> > - **Effect of slot size.** Because using just $M$ (usually 8 or 16) dimensions in IODINE’s slots can deviate too much with respect to the original slot-size (64) used in the IODINE paper, we provide additional results in which we train IODINE with a wide range of other slot-sizes: 8, 16, 32, 64, 128 and 256.  We find that all these settings still perform significantly worse than our model.
> >
> > See these new results in the Fig. 6 of our revised submission or (alternatively) at this anonymized link: [https://imgur.com/a/ULxlnT0](https://imgur.com/a/ULxlnT0).
> >
> > > Empirical result showing fewer active dimensions in IODINE compared to SLATE/Slot Attention was also shown in Dang-Nhu (2021).
> > >
> >
> > Thank you for pointing this. In the revised submission, we have now cited Dang-Nhu (2021) in Section 6.1 regarding this effect of VAE prior.
> >
> > > Letter $K$ used twice to mean different things: number of prototypes and number of ground-truth factors of object variation.
> > >
> >
> > Thank you for pointing. In the revised submission, we have changed the letter notation for the number of ground-truth factors of object variation.

---

> > > ### Comment · Reviewer_NtVS · 2022-11-16
> > > **Answer - score increase**
> > >
> > > I am very satisfied with all the improvements and clarifications made by the authors. Besides, I would like to emphasize that
> > > - evaluating on the CLEVR synthetic dataset is consistent with many recent contributions in the same field and it's already quite a complex dataset, especially the textured version.
> > > -  the contribution is addressing a crucial challenge in representation learning which is the main focus of ICLR. The ability to reliably perform intra-slot disentanglement between blocks will bring major benefits for structured scene manipulation and style transfer. As an industry practitioner I expect that this will be useful for a wide range of product applications.
> > >
> > > **In the current state I am absolutely confident that this submission deserves acceptance and I have increased  my score to 8.**

---

> > > > ### Author Response · Authors · 2022-11-16
> > > > **Score Increase**
> > > >
> > > > We sincerely thank you for the strong support of our paper. We are glad to hear that we addressed your concerns satisfactorily and we also completely agree with your points.

---

> > > > > ### Comment · Reviewer_NtVS · 2022-11-21
> > > > > **Additional comment**
> > > > >
> > > > > Since there is a consensus on acceptance I still have a comment for a potential camera-ready version. Concerning the statement that “Single-dimension representation of a factor is difficult to use as input to a Transformer". A real number can be canonically mapped to a vector of dimension 1. I agree that the input dimension of transformers is typically larger, but in the context of sequential data modelling I have seen successful applications of transformers encoders to digital ink where the input dimension is 3 or 4 [1], therefore I would be careful with such strong claims that dimension 1 wouldn't work. Maybe you can have a more cautious phrasing of this?
> > > > >
> > > > > [1] COSE: compositional stroke embeddings, Aksan et al., NeurIPS 2020

---

> > > > > > ### Author Response · Authors · 2022-11-25
> > > > > > **Reply**
> > > > > >
> > > > > > Thank you for the comment. We agree with you that a more cautious phrasing would be better. We will incorporate this in the final version.

---

### Official Review · Reviewer_v6kB · 2022-10-24

**Confidence:** 5
**Correctness:** 4
**Technical Novelty And Significance:** 4
**Empirical Novelty And Significance:** 4
**Recommendation:** 8

**Clarity, Quality, Novelty And Reproducibility:**

* See above, very high clarity.
* See above, great quality.
* Great novelty: the group-RNNs + Concept Memory choice + disentangling metric modification + Block Binding were all novel to me.

**Strength And Weaknesses:**


Overall, I really liked this paper in this current form so I have very little extra to request:


1. The Abstract and Introduction are very strong, easy to follow yet entirely clear, it was really well done. Figure 1 conveys nearly the entirety of the model in one single diagram which is fantastic.
   1. The only nitpick I would have is to indicate that there are M blocks in total, this is the only missing symbol currently.
2. The Concept Memory bottleneck is potentially both the biggest novelty or drawback of the technique in its current form.
   1. It was surprising to me that you would directly attend into it by multiplying s_nm with Cm. Somehow I would have expected something more akin to a query/key setup with additional Linear layers. Did you try something like that?
   2. The choice of making the prototypes the direct outputs is rather strong.
      1. I feel like this may be too constraining and brittle, for example in complex situations?
      2. I had to find how many prototypes K you used in the Appendix. You seem to use K=64 throughout, which is rather small. This seems rather much more constraining than what is done VQ-VAE/GAN or GroupVIT-style models.
      3. How “delta-ish” were the softmax attention in general? If the softmax becomes extremely peaky, this means you considerably restrict the “space” of representation (i.e. in the limit, you would only represent 64 possible values, given your choice of K)
      4. What happens if the dictionary size is much larger? What about if you share it between groups?
      5. Did you try outputting the “distance” to the prototypes instead, to have more flexibility and expressive power?
   3. Didn’t you have to use more flexible attention mechanisms, ala Gumbel, to train these prototypes well enough? I would expect the gradients to die off rather quickly in unused prototypes.
3. The Block Binding is a very interesting choice, and it seems rather useful to me so I will probably give it a try in my own work.
   1. However, I would have expected you to try the simplest choice of “just” concatenating all the groups together per slot first? I.e. provide $S=\lbrace s_n \rbrace$, where $s_n = Concat_m(s_{n, m})$.
   2. You never compared against that, even in Section 6.2? I would never have expected “the bag of blocks” to be an appropriate representation, provided as a set to the decoder. You do have the block-slots directly, so why don’t you just keep that structure? How does Figure 6a change with that choice?
4. The results are strong, and the improvement to the metric is well executed. Figure 2 is clear and very impactful.
5. Figure 4 and the methodology for obtaining it is not clear enough.
   1. How did you “mask out” part of the images? Did you use the attention weights?
   2. The corresponding section on 6.1 at the end of Page 7 needs to be expanded to provide more details, as this is a very interesting assessment which is the only way to assess what the groups and prototypes really learn.
   3. How many prototypes end up being learnt/used? Can we get some kind of “coverage” metric for their relative importance / specialization?
6. Figure 5 is strong and very interesting to see.
   1. It might be worth flagging that the model gets the reflection right (e.g. last row, second to last column on CLEVR-Hard). Do you know if this is thanks to your model, or “simply” due to the Transformer+dVAE decoder being really good at handling these?
   2. How exactly did you select the groups? You say it was easy to identify and select them, but you do not precisely say how.
7. Even though you target more complicated versions of CLEVR, this is still “just” CLEVR, which is rather easy to get results on.
   1. Did you try targeting Kubric [1] (MOVi-C onwards, discarding the temporal video aspect) or other more complex yet still manageable datasets?


[1] https://arxiv.org/abs/2203.03570

**Summary Of The Paper:**

This paper presents a modification of Slot Attention to make it learn sub-components which are independent of each other (blocks), built out of prototypes which are attended to be independent RNNs. The model is trained through reconstruction (with a transformer+dVAE decoder like recent works) and performs very well on several extensions of the CLEVR dataset, including one with textures (current techniques struggle with non-trivial textured datasets). The results are strong and the presentation is very clear.

**Summary Of The Review:**

Overall, I found this paper extremely interesting, presenting several key novel ideas which appear simple in hindsight which is always a great sign. Results are strong, the evaluation is good with good ablations and the paper demonstrates what I wanted to see and understand.

As a practitioner of this field, this is absolutely something I will try to leverage and build upon, hence I would recommend for acceptance.

---

> ### Author Response · Authors · 2022-11-15
> **Response to Reviewer v6kB (1/2)**
>
> Thank you for this insightful and positive review! The comments and the suggested experiments are extremely helpful.
>
> > In Figure 1, indicate that there are $M$ blocks in total.
> >
>
> Thank you for this suggestion! We have now updated the figure to indicate that there are $M$ blocks per slot.
>
> > How peaky are the softmax attention weights over the prototypes?
> >
>
> We visualize the softmax attention weights over the prototypes in Fig. 8 in the appendix (see a screenshot of this figure at the anonymous link: [https://i.imgur.com/sewhAKL.png](https://i.imgur.com/sewhAKL.png)). We note that the attention weights are not one-hot. **We also do not use any temperature control to make the softmax peaky.** Rather the weights are distributed softly over several prototypes. Thus, the retrieved representation is not a ‘hard’ selection of a single prototype but rather a soft linear combination of multiple prototypes.
>
> > Could $K=64$ prototypes be too few and limiting? What if the dictionary size is much smaller or larger?
> >
>
> Because the retrieved representation is a soft linear combination and not a hard selection, having 64 prototypes does not mean only 64 possible outcomes but has much greater flexibility. We find evidence of this in Fig. 6(b) where we test the performance of our model with different number of prototypes: $K=8$, $64$, and $1024$. We find that the informativeness score of the slots is not affected by these choices of $K$.
>
> > What happens if we share the concept memory across blocks?
> >
>
> We thank you for suggesting this interesting experiment. We tested this by making a variant of our model in which we share both the RNN parameters and the concept memories across all $M$ blocks. We added this result in Fig. 7 in the revision and a screenshot is attached at this anonymized link: [https://imgur.com/a/8JGBlbb](https://imgur.com/a/8JGBlbb). We can see that as the level of weight-sharing increases, the completeness score decreases. In other words, with more weight-sharing among blocks, the model tends to use more blocks on average to represent a single factor. However, as this is not a drastic degradation, we can say that weight separation is not the major factor responsible for the emergent disentanglement. What is crucial for getting good disentanglement is doing each block’s prototype attention and refinement independently.
>
> > Did you try outputting the distance to the prototypes instead?
> >
>
> Thank you, the suggestion sounds interesting. We will consider this experiment in the final version.
>
> > Did you try concatenating the blocks rather than block-binding?
> >
>
> We tested this variant and added the result in Fig. 7 in the appendix (see screenshot at the anonymized link: [https://imgur.com/a/8JGBlbb](https://imgur.com/a/8JGBlbb)). This variant produces noticeably worse disentanglement and completeness scores and shows that the proposed block-binding approach is a more optimal choice.
>
> > Methodology for obtaining Fig. 4 is not clear enough.
> >
>
> Thank you for pointing. In the revised submission, in appendix Section E.1, we have now formally described the process of obtaining Fig. 4.
>
> > Did you use attention weights to mask out the images in Fig. 4?
> >
>
> Yes, we used the attention weights to mask out the images and visualize the object represented by a block.
>
> > How many prototypes get used? Do gradients die off quickly in unused prototypes?
> >
>
> In the revised submission, in Fig. 9 in the appendix, we visualized the average attention weight received by each prototype over a large batch of 300 images (see a screenshot of this figure at the anonymous link: [https://i.imgur.com/0rYJ5h2.png](https://i.imgur.com/0rYJ5h2.png)). This is meant to visualize how much each prototype gets used and whether a prototype is dead. We can see that a large majority of prototypes have a non-zero attention weight and are thus ‘active’ while the ‘dead’ prototypes are much fewer.
>
> > In Fig. 5, are correct reflections due to the transformer decoder or due to the proposed block-slot attention?
> >
>
> We think it’s both. That is, the block-slot representation provides a good form of structured representation that the transformer decoder can easily utilize to generate correct and coherent scenes.
>
> > In Fig. 5, how did you select the blocks for property swapping?
> >
>
> We selected them by first applying $K$-means on each $m$-th block as in the qualitative analysis of Section 6.1. and Fig. 4. We then manually inspected the resulting object clusters and identified which blocks are responsible for representing a specific property. We added this description in our revision in the appendix.

---

> > ### Author Response · Authors · 2022-11-15
> > **Response to Reviewer v6kB (2/2)**
> >
> > > Did you target Kubric or MOVi-C/D/E datasets?
> > >
> >
> > Thank you for this suggestion. While we did not directly target the highly complex MOVi-C/D/E that contains real-world objects from Google Scanned Objects (GSO) dataset. However, we made a variant of the CLEVR dataset containing a subset of the Google Scanned Objects (termed the CLEVR-GSO dataset). At this anonymous link: [https://imgur.com/a/y1jRjRa](https://imgur.com/a/y1jRjRa), we provide some sample images of this dataset and a preliminary result where we visualize the $K$-means clusters on various blocks learned by our model (analogous to Fig. 4). Here, we can see the emergence of concepts like color, object-type, orientation, and position. While this is preliminary, however, for the final version, we will consider evaluating this more thoroughly.
> >
> > Motivated by the promising result on the CLEVR-Tex and the CLEVR-GSO dataset, going further to more complex images such as Kubric will be one of our main future work.

---

> > > ### Comment · Reviewer_v6kB · 2022-11-17
> > > **Answer**
> > >
> > > Thanks a lot for the this great rebuttal and the additional results and precisions in the Appendix.
> > > They cover all my requests and original questions, thanks!
> > >
> > > This is still a strong accept in my case and I would recommend this paper for ICLR.

---

> > > > ### Author Response · Authors · 2022-11-18
> > > > **Reply**
> > > >
> > > > We are pleased to hear this. We sincerely thank you for your continued strong support of our work.

---

### Official Review · Reviewer_jDiA · 2022-10-29

**Confidence:** 4
**Correctness:** 4
**Technical Novelty And Significance:** 3
**Empirical Novelty And Significance:** 3
**Recommendation:** 6

**Clarity, Quality, Novelty And Reproducibility:**

I wrote above my concerns.


**Strength And Weaknesses:**

**Strengths:**

(1) Overall, this paper is well written, and the technical details are easy to follow.

(2) The main idea of representing a factor of object variation, such as color, texture, and position, is nice and appealing.

(3) The main contribution of this paper is a new method for binding block factors within a slot.

(4) The experiment results support the proposed approach, including the evaluation of the DCI framework, which is a valuable addition to the paper.


**Weaknesses:**

**Experiments.** I am mostly concerned with the fact that this approach is mainly proved to be working on synthetic datasets. Thus, I wonder how it would generalize to a real domain. I think the idea is appealing, but I am honestly not sure it will work on real-life datasets.


**Novelty.** First of all, I would like to note that the main idea of this paper maintains the high standards of the conference. However, the authors should also know that I had indeed considered this idea important and implemented it 14 months ago, when the first slot-attention paper was published. The method of disentangling the structure within a slot in order to represent a factor of four object variations, such as color, shape, texture, and position, appears to be working entirely on synthetic datasets (CLEVR-like). Due to the fact that it did not work on real domains, as well as the simple nature of the contribution of the main idea (which is, to me, way too simple in this case, although there is a compelling story behind it), I decided to abandon this research direction. Despite acknowledging the importance of this paper, we should also be aware that its contributions could not be substantial enough, especially if we look at the results on real domains.

**Summary Of The Paper:**

This paper presents a novel object-centric representation called Block-Slot Representation, which, unlike the conventional slot representation, provides concept-level disentanglement within a slot, such as color, texture, and position. In comparison with the previous methods, the approach demonstrated significantly better disentanglement of object properties, including complex textured scenes. Finally, the proposed method improves the results on a number of synthetic datasets (CLEVR).


**Summary Of The Review:**

I wrote above my concerns. I am open to the authors' feedback and other reviewers' opinions.

---

> ### Author Response · Authors · 2022-11-15
> **Response to Reviewer jDiA**
>
> We thank you for your comments and the positive recommendation!
>
> > Evaluation on real scene images.
> >
>
> We respond to this point at length in the common answers above.
>
> > Simple nature of the contribution.
> >
>
> We respectfully disagree. Seemingly, realizing disentanglement in slot representations might look like a straightforward direction. However, as acknowledged by the other reviewers, our method contains various novel ideas. We do not take the usual path of introducing the VAE KL term to achieve dimension-wise disentanglement. We motivate why we need deterministic and vector-based disentangled property representation and show how it can be realized. In doing so, we also introduce several novel ideas, such as independent group RNNs, concept memory bottleneck, a new block binding method, and new methods for visualization and analysis. We agree with Reviewer v6kB who pointed out that these are novel ideas.

---

> > ### Comment · Reviewer_jDiA · 2022-11-17
> > **Response**
> >
> > After reading the authors' feedback and other reviewers' opinions, I would like to thank the authors for their rebuttal. In general, I am pleased with all the improvements and clarifications that the authors have made. The rebuttal addresses most of my concerns. I am leaning towards acceptance of the paper since it maintains the high bar of the conference quality. I vote for 6.

---

> > > ### Author Response · Authors · 2022-11-18
> > > **Response**
> > >
> > > > Pleased with improvements and clarifications. Paper maintains the high bar of the conference quality
> > >
> > > We are pleased to hear this and we are thankful for your continued support.

---

### Author Response · Authors · 2022-11-15
**List of Revisions**

1. Added more analysis of IODINE (see Fig. 6 or its screenshot via the anonymized link: [https://imgur.com/a/ULxlnT0](https://imgur.com/a/ULxlnT0)).
    1. an analysis of the effect of $\beta$ on IODINE
    2. an analysis of the effect of slot size on IODINE’s DCI scores.
2. Extended the analysis of our model by evaluating the following variants (see Fig. 7 or its screenshot via the following anonymized link: [https://imgur.com/a/8JGBlbb](https://imgur.com/a/8JGBlbb))
    1. Concatenating the blocks to construct a slot instead of block-binding when providing blocks to the transformer decoder.
    2. Sharing the concept memory across blocks instead of having a separate memory per block.
    3. Applying query, key, and value projections in the concept-memory attention.
3. In Section 6.2, highlighted the correct object reflections in the generated images after swapping the object properties.
4. Added a more detailed description in the appendix of how the visualization of $K$-means clusters of Fig. 4 was generated.
5. Added a reference to Dang-Nhu (2021)’s result about the effect of the VAE prior on DCI scores. Also, fixed the description of Dang-Nhu (2021) in the related work.
6. Added a visualization in the appendix showing the prototype usage/coverage over a large batch of images.
7. Added a table (Table 1 in appendix) comparing the computational needs of our model with IODINE.
8. Highlighted that we will release our code and all the datasets through a ‘Reproducibility Statement’.
9. To make space in the main paper to address the reviewer comments, made the description of DCI in Section 4 more concise and moved the discussion of the limitations and the future work to Appendix F.

---

### Author Response · Authors · 2022-11-15
**Common Response**

We thank all the reviewers for providing a thorough and insightful review! We are encouraged by the positive recommendations and comments such as **well-written**, **easy-to-follow, main idea is nice and appealing, extremely interesting, really liked this paper, maintains high-standards of the conference, results are strong, emergence is very impressive.** In this reply, we address some common concerns.

> Method shown on synthetic datasets but not on real datasets or more challenging benchmarks. [Reviewers *jDiA and HPVD*]

While we agree that dealing with real images is a desired end goal, the contributions of our paper should be viewed in the context of where the state-of-the-art stands today. In the **fully-unsupervised regime**, the community still does not have a scene decomposition method that works reliably well on complex real scenes. The good news is that there has been considerable progress in the last several years — driven mainly by synthetic datasets of gradually increasing complexity. This is very promising progress. As such, all our datasets, although synthetic, are either comparable or more complex than previous works **that pursued *intra-slot disentanglement***. The **CLEVR-Tex dataset actually takes a big step forward in textural complexity in this line of research.** With previous models failing, the fact that our models show success on this dataset expands the scope of what unsupervised models are now capable of doing. This, we believe, is something worth sharing with the community.

> What would happen if linear projections were used for computing queries, keys, and values in concept-memory attention? [Reviewers *v6kB and NtVS*]


This is an interesting suggestion. We tested this variant by applying linear projections to compute queries, keys and values for concept-memory attention. We have added the results in Figure 7 in the appendix of the revised submission. It can be also viewed at this anonymized link: [https://imgur.com/a/8JGBlbb](https://imgur.com/a/8JGBlbb). We found this variant to be less stable than our default model. We also see that the segmentation quality becomes noticeably worse. We hypothesize that this is because the learned linear projections interfere with the convergence of the slot-refinement iterations. Furthermore, we see a worsening of the DCI performance as well.

---

### Decision · Program_Chairs · 2023-01-20

**Decision:**

Accept: poster

**Justification For Why Not Higher Score:**

The paper didn't attempt to push the limits in terms of the data complexity that this class of models can address and only evaluates on fairly simplistic synthetic multi-object scenes, while other recent works have made attempts to bridge the gap to naturalistic and real-world scenes. It is unclear how the within-slot decomposition mechanism proposed in this paper would scale to more complex data. Nonetheless, the paper is of very high quality and presents a solid and elegant contribution of significant interest to the community.

**Justification For Why Not Lower Score:**

All reviewers agree that this is a strong paper worthy of being accepted at the conference.

**Metareview: Summary, Strengths And Weaknesses:**

This paper addresses the problem of object discovery from synthetic multi-object scenes. It introduces an architecture called Block-Slot Attention, which extends prior work on Slot Attention [1] by introducing a further decomposition mechanism that decomposes individual object slots into so-called blocks. The resulting representation is referred to as block-slots. The authors demonstrate that this simple inductive bias results in emergent decomposition of object properties into particular blocks, e.g. one block is responsible for color, one for shape etc. – the model is validated on synthetic multi-object scenes of varying visual complexity.

All reviewers agree that this is a strong submission: it is very well-written, well-structured and overall very clear. The method is simple and appealing. The reviewers highlighted that the experiments are strong and convincing, and also mentioned that the particular choice (and improvement of) the metric for evaluation is well-executed.

The main concerns by the reviewers revolve around the simplicity of the chosen datasets for evaluation: they are all variants of the CLEVR dataset. Although the authors did include a textured version of the dataset, CLEVRTex, that has significantly higher visual complexity, no results on naturalistic or real-world datasets are presented. It is unclear whether the proposed block-slot decomposition would generalize or show benefits on more complex real-world datasets. This, however, can be excused since the community is still struggling overall to make this class of models reliably generalize to more naturalistic or real-world data, although recent progress has been made. Other minor concerns around ablations and compute requirements were well-addressed during the rebuttal.

Overall, this is a high-quality paper of significant interest to the community for which I can clearly recommend acceptance.

[1] Locatello et al., Object-Centric Learning with Slot Attention (NeurIPS 2020)


**Note From Pc:**

if the above contains the word "oral" or "spotlight" please see: "oral" presentation means -> notable-top-5% and "spotlight" means -> notable-top-25%. As stated in our emails, we are disassociating presentation type from AC recommendations